# Computational characterization of the xanthan gum glycosyltransferase GumK

Davide Luciano[1], F. Emil Thomasen[2], Kresten Lindorff-Larsen[2], Gaston Courtade[1]*

**1** Department of Biotechnology and Food Science, NTNU Norwegian University of Science and Technology, Trondheim, Norway, **2** Linderstrøm-Lang Centre for Protein Science, Department of Biology, University of Copenhagen, Copenhagen N, Denmark

* gaston.courtade@ntnu.no

## Abstract

The activity of GT-B glycosyltransferases (GTs) depends on their conformational flexibility and high substrate specificity, but the molecular basis of these features is still not well defined. The GT70 family contains a single well-characterized enzyme, GumK, a glucuronosyltransferase from *Xanthomonas campestris* required for xanthan gum biosynthesis. Here, we applied multiscale molecular simulations and sequence analysis to probe GumK dynamics and substrate specificity. We show that GumK undergoes twisting and bending motions constrained by interdomain contacts and modulated by membrane anchoring. Acceptor-substrate binding within an amphiphilic clamp promotes opening, whereas donor-substrate binding stabilizes closure, defining a substrate-dependent catalytic cycle. Specificity for UDP-glucuronate is mediated by a conserved electrostatic environment centered on Lys307 and a hydrophobic triad that orients the sugar moiety. On the acceptor side, the binding site selectively accommodates polyisoprenyl carriers up to three isoprene units in length and wraps around the substrate, constraining the trisaccharide moiety in a catalytically competent conformation. Comparative analysis highlights GumK-specific motifs that distinguish it from homologous GTs. This work provides mechanistic insight into the GT70 family and the dynamic behavior of GT-B enzymes, establishing principles for the rational engineering of GumK to modify the monosaccharide composition of xanthan gum.

## Author summary

Glycosyltransferases are enzymes that build many of the sugars and polysaccharides essential for life. One of them, called GumK, is responsible for a key step in producing xanthan gum—a natural polymer widely used as a thickener in food and industrial materials. Despite its importance, how GumK works at the molecular level has remained unclear. In our study, we used multiscale computer simulation

**Data availability statement:** Data and scripts used for running and analyzing simulations, and for generating the figures are available from https://github.com/gcourtade/papers/tree/master/2025/GumK.

**Funding:** G.C. and D.L. were funded by the Novo Nordisk Foundation (url: https://novonordiskfonden.dk/en/; grant number NNF18OC0032242 awarded to G.C.). D.L. received salary from the Novo Nordisk Foundation. The funder had no role in study design, data collection and analysis, decision to publish, or preparation of the manuscript.

**Competing interests:** The authors have declared that no competing interests exist.

strategies to explore the dynamics of this enzyme and how it interacts with the cell membrane and its sugar substrates. We found that GumK acts like a flexible clamp that opens and closes as it binds two different sugar molecules through specific residues that may control the enzyme's selectivity. We also discovered how the surrounding membrane helps the enzyme remain in the correct orientation to perform its function. By identifying the molecular details that determine GumK's selectivity and flexibility, our work provides a foundation for modifying this enzyme to produce xanthan gum with new properties. More broadly, our findings help explain how a large family of related enzymes controls sugar transfer in living organisms, with potential applications in biotechnology and materials science.

## Introduction

Glycosyltransferases (GTs) are key enzymes involved in the biosynthesis of many biologically relevant molecules. The expression of specific subsets of these proteins can profoundly impact an organism's phenotype, influencing cellular processes such as signaling, immune response, and structural integrity. GTs glycosylate nucleic acids, proteins, and small molecules, and are essential for the biosynthesis of oligosaccharides and polysaccharides, which play crucial roles in metabolism and cellular interactions [1]. Due to their broad biochemical capabilities, GTs possess significant industrial and pharmaceutical potential. They can be harnessed as biocatalysts for the synthesis of novel materials and bioactive compounds, or targeted by drugs to interfere with pathogenic biosynthetic pathways [2–4]. Despite their importance, the activity of these enzymes is still not fully characterized [3,5–7].

GTs exhibit remarkable structural and functional diversity, encompassing multiple fold types, including GT-A, GT-B, GT-C, and lysozyme-like architectures [8,9]. This structural variability is reflected in their catalytic mechanisms, which involve the transfer of a monosaccharide unit from an activated donor substrate, typically a nucleotide sugar, to an acceptor molecule. Depending on the reaction mechanism, GTs can follow three distinct catalytic pathways and are classified as either inverting or retaining enzymes, depending on whether they reverse or preserve the anomeric carbon configuration of the donor substrate [10]. The key scientific challenge is to understand and predict how these proteins select their donor and acceptor substrates, as this has a major impact on both their biological function and their industrial applications.

Out of the 138 glycosyltransferase families listed in the CAZy database [11], many remain poorly characterized due to challenges in protein expression and substrate instability. These limitations make it difficult to study their biochemical properties, such as substrate specificity and selectivity, thereby constraining our ability to exploit

them in industrial and biotechnological applications. Among these underexplored families, GT70 has a single well-characterized member, GumK, which plays a key role in the biosynthesis of xanthan gum, a widely used polysaccharide produced by *Xanthomonas campestris*.

The enzymatic activity of GumK has been extensively investigated, and several X-ray crystal structures are available in the Protein Data Bank (PDB) [12]. Interest in this enzyme arises from its role in the biosynthesis of xanthan gum, one of the most versatile and widely used polysaccharides in the materials industry [13–15]. Given its broad range of applications, both chemical and biochemical strategies have been explored to enhance its functionality [16]. A deeper understanding of GumK's substrate selectivity could further advance these efforts by enabling the tuning of its specificity and the incorporation of alternative sugar units into the polysaccharide. Such modifications could yield novel xanthan derivatives with tailored mechanical and functional properties.

The crystal structure of GumK (PDB ID: 2HY7) adopts a typical GT-B fold, with both the N- and C-termini located within the same domain [12]. The protein consists of two Rossmann-like domains, conventionally referred to as the N-domain (residues 13–201 and 362–380) and the C-domain (residues 210–361), connected by a flexible linker (residues 202–209). The C-domain binds UDP-glucuronate (GlcA-UDP) as the donor substrate, whereas the N-domain interacts with the membrane-anchored acceptor substrate, $\alpha$-D-Man-(1→3)-$\beta$-D-Glc-(1→4)-$\alpha$-D-Glc-1-diphospho-ditrans,polycis-undecaprenol (Man-Cel-UNDPP). This linker provides structural flexibility, enabling domain movements that are essential for enzymatic function [12] (Fig 1).

Unlike other enzyme folds, where the catalytic site is a well-defined pocket, GumK's active site is located within the cleft between the two domains [17], making its precise conformation highly dependent on interdomain motions. GumK follows an inverting catalytic mechanism, in which a nucleophilic substitution (SN2-like) reaction transfers the sugar moiety with inversion of the anomeric configuration. The catalytic residue responsible for this reaction is Asp157, located in the acceptor domain, which activates the acceptor substrate and facilitates the reaction through nucleophilic displacement [12]. The conformational flexibility of the linkers between the two domains enables two major types of motion—twisting and bending—which have been observed in computational studies [18]. These motions are believed to be essential for enzymatic activity, facilitating substrate binding and proper positioning for catalysis [17]. However, the full conformational landscape of GumK remains unknown. Identifying the range of possible domain orientations is crucial for defining active conformations, understanding substrate binding and specificity, and determining how these dynamic processes influence reaction kinetics [6]. It also contributes to a broader understanding of how glycosyltransferases achieve substrate selectivity and catalysis.

While experimental studies have provided critical structural data, computational methods now complement them by adding atomistic detail to protein dynamics and enzyme activity, generating ensembles of conformations consistent with experimental observations. Among the various computational approaches, molecular dynamics simulations can identify key conformations, reveal hidden molecular interactions, and refine our understanding of how structural flexibility influences function and activity. Ultimately, these insights can guide the design of targeted experiments, enhancing our ability to manipulate and engineer these enzymes for both fundamental research and industrial applications.

Studying GumK computationally is challenging due to the high flexibility of both the protein and its substrates, its interaction with a membrane-embedded acceptor, and the poorly defined catalytic site located within the interdomain cleft. This flexibility gives rise to a complex conformational landscape, complicating the characterization of the structural dynamics that regulate its function and substrate recognition.

To address these challenges, we applied a multi-tiered computational approach that combined normal mode analysis (NMA), unbiased molecular dynamics simulations, coarse-grained OPES simulations, and Hamiltonian Replica Exchange (HREX) for enhanced sampling. This strategy provides new insights into the flexibility, substrate interactions, and catalytic site dynamics of GumK. Our findings lay the foundation for future experimental validation and potential biotechnological applications.

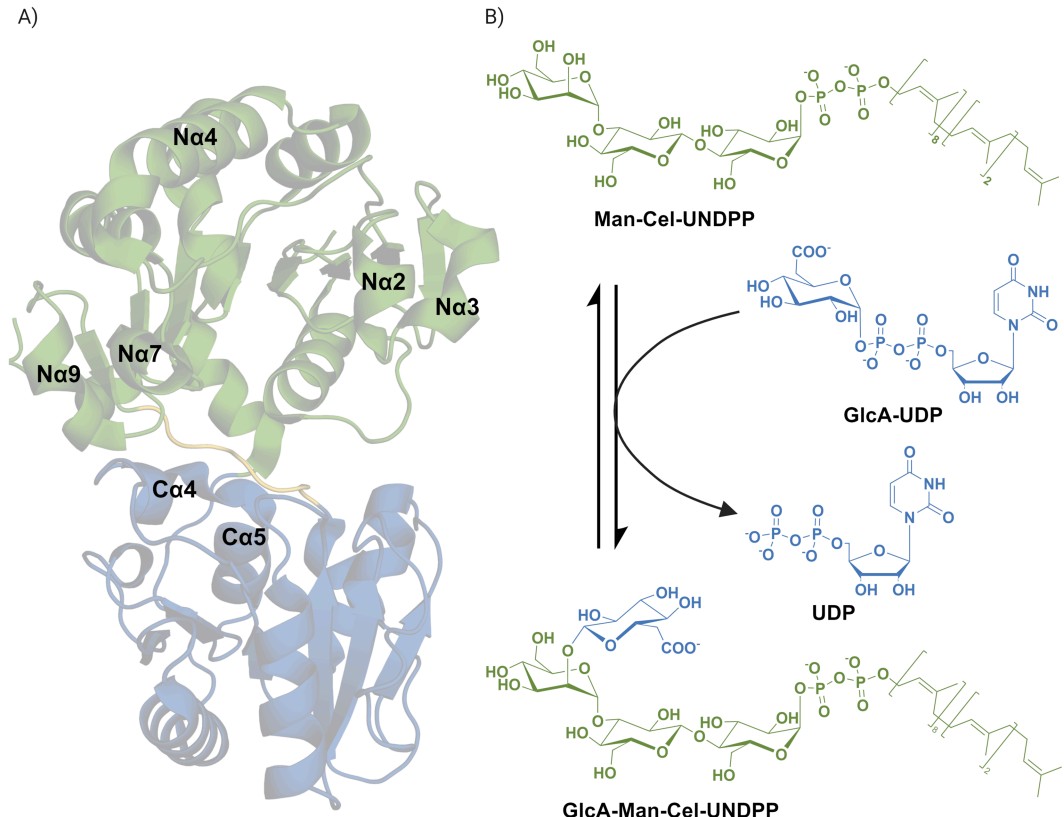

**Fig 1**. **A) Crystal structure of GumK (PDB: 2HY7), colored according to domain nomenclature: the N-domain in green, the C-domain in blue, and the linker in yellow.** Key regions of the protein discussed in the main text are labeled. B) Reaction catalyzed by GumK, with colors corresponding to the domain assignments. The acceptor substrate, Man–Cel–UNDPP, binds to the N-domain, while the donor substrate, glucuronate–UDP (GlcA–UDP), binds to the C-domain. The enzyme catalyzes the formation of a $\beta$-(1,2)-glycosidic bond between glucuronate and mannose in the acceptor substrate, producing GlcA–Man–UNDPP.

## Materials and methods

We used molecular dynamics simulations to characterize GumK's structural behavior and interactions with native substrates (Table 1). Below, we describe each simulation in detail.

### Normal mode analysis and ClustENMD

Normal mode analysis of the GumK crystal structure (PDB ID: 2HY7) was performed using the anisotropic network model (ANM) implemented in the Python library ProDy [19]. The Hessian matrix of the elastic network was constructed using default parameters, namely a cutoff distance of 15 Å and a spring constant of 1.0. The resulting matrix was then diagonalized to obtain the first 20 normal modes.

The first two normal modes were used to define the deformation vectors in ClustENMD to sample full-atomistic conformations of GumK. The target RMSD at each generation was set to 1.0 Å, with a total of 50 generations. The number of clusters per generation followed a linear distribution, ranging from 10 to 510 in increments of 10. The GumK crystal structure (PDB ID: 2HY7) was described using an implicit solvent model, and at the end of each generation, the system was heated to a target temperature of 300 K and energy-minimized using the default ClustENMD force field, AMBER99SB-ILDN [20].

**Table 1**. Summary of the different simulations performed in this work.

| Simulation | System | Force field | Time | Objective |
|---|---|---|---|---|
| ClustENMD | GumK free in solution | AMBER99SB-ILDN | 50 gens | Conformational sampling |
| Unbiased MD | GumK + membrane | CHARMM36m | 1 $\mu$s | Conformational sampling |
| OPES | GumK + membrane | MARTINI3 rescaled | 10 $\mu$s | Membrane docking |
| HREX | GumK C-domain + glucuronate-UDP | CHARMM36m | 1 $\mu$s | Sugar binding mode sampling |
| HREX | GumK C-domain + glucose-UDP | CHARMM36m | 1 $\mu$s | Sugar binding mode sampling |
| Unbiased MD | GumK + membrane + acceptor substrate | CHARMM36m | 1 $\mu$s | Conformational sampling |

### Coarse-grained OPES metadynamics MD simulation

We used an OPES simulation to study the interaction of GumK with the membrane. The MARTINI3 force field was employed with rescaled protein–protein interactions to ensure a stable docked complex within the membrane [39]. The coarse-grained model of the full-length AlphaFold-predicted GumK structure (UniProt ID: Q8GCH2) was generated using the `martinize.py` Python script. The complete system included a membrane with a phospholipid composition as close as possible to that described in the next section. However, due to the lack of parameters for cardiolipin, this lipid was replaced by additional phosphatidylglycerol molecules. The final membrane composition consisted of 1,2-dilauroyl-*sn*-glycero-3-phosphoethanolamine (DLPE) and 1,2-dilauroyl-*sn*-glycero-3-phospho-(1'-rac-glycerol) (DLPG) in a 1.2:1 ratio. System neutralization at an ionic strength of 0.1 M was performed using the `insane.py` Python script. The MARTINI elastic network was designed to maintain the two domains in a closed conformation while allowing free movement of the termini. The starting conformation was placed beneath the membrane in an unbound state.

The system was first subjected to energy minimization, followed by equilibration in the NPT ensemble in two stages, progressively increasing the integration time step. In the first equilibration stage, a time step of 2 fs was used for a total of 1 ns. The second equilibration, also in the NPT ensemble, was run for 10 ns with a time step of 20 fs. After these two equilibration steps, an unbiased simulation was performed for 500 ns. During the second equilibration and the subsequent 500 ns production run, a cylindrical-shaped potential was activated to allow the system to equilibrate in the presence of the new potential. The cylindrical-restrained OPES simulation was then performed with a time step of 20 fs in the NPT ensemble for 5 $\mu$s. Throughout all simulations, the temperature was maintained at 300 K using the V-rescale thermostat with a coupling constant of 1.0 ps. The pressure was kept at 1 bar using the Parrinello–Rahman barostat, with a coupling constant of 12 ps and a compressibility of $3.4 \times 10^{-4}$ during the OPES simulation.

The backbone bead of Leu20 in the acceptor domain was used as a reference atom to constrain GumK within a cylindrical-shaped restraint potential, manually defined to prevent the protein from freely diffusing in solution. Three collective variables (CVs) were biased using different energy barriers: (1) the projection of the reference atom in the acceptor domain along the cylinder's main axis; (2) the cosine of the angle describing the tilt of the acceptor domain relative to the membrane (S1D Fig); and (3) the number of contacts between the membrane and the most hydrophobic region surrounding Trp98. The cosine function was employed to prevent undesired sampling of protein rotation relative to the membrane's main axis, ensuring that the tilt angle remained within the range of 0° to 180°.

All simulations were performed using GROMACS 2022.3 [21] in combination with PLUMED 2.8 [22]. Using the OPES bias, the free energy surface (FES) as a function of these collective variables was computed through a reweighting method [23] (Fig 2A). To assess the convergence of the simulation, we analyzed the evolution of the CVs over 5 $\mu$s. The results show that multiple binding events were sampled throughout the trajectory, indicating that the system explored a range of conformational states with several transitions between them (S1A,B Fig).

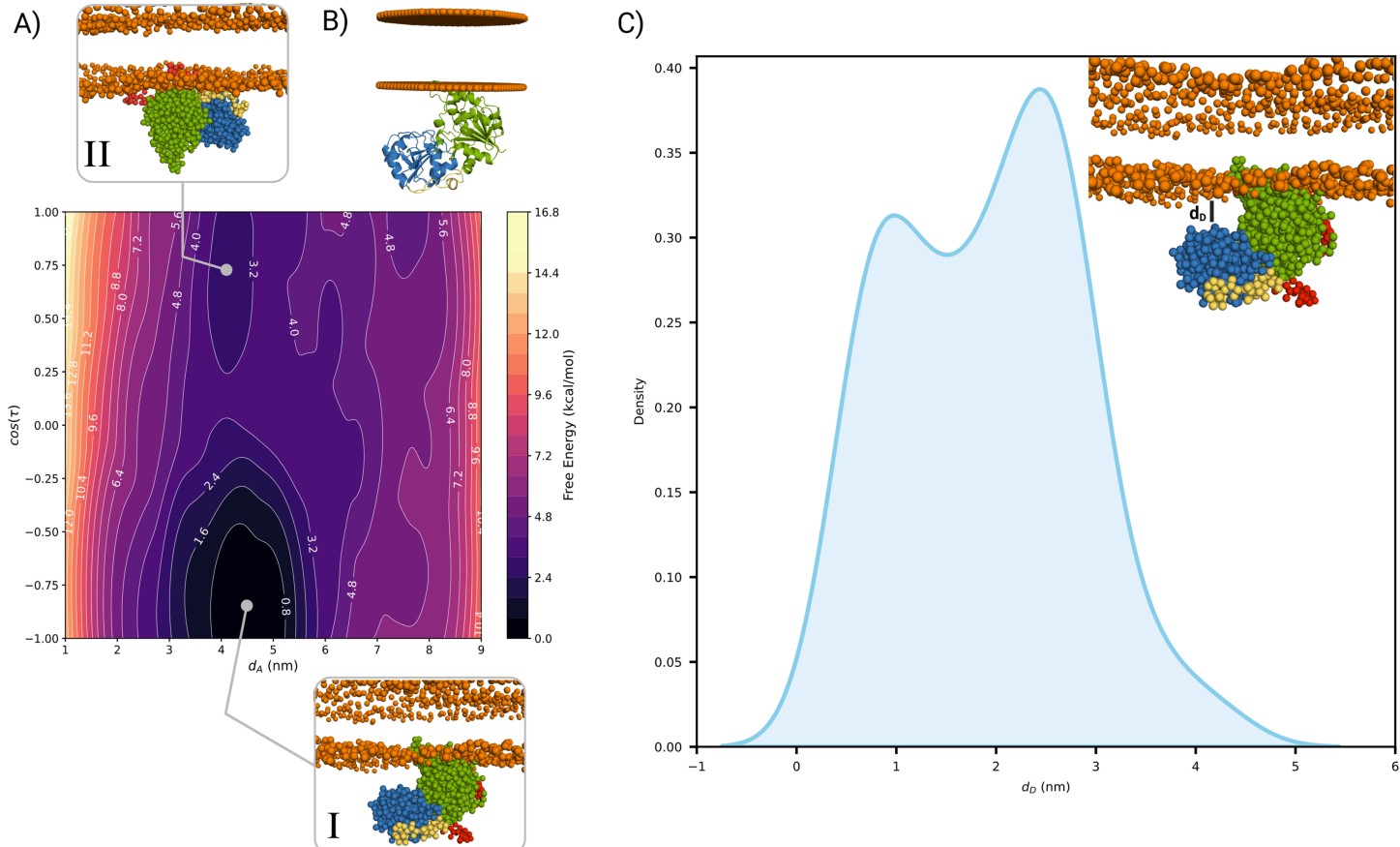

**Fig 2. A) Free-energy surface (FES) from the cylindrical-restrained OPES simulation with the MARTINI force field, plotted against the two biased collective variables: the acceptor-domain distance from the membrane ($d_A$) and the tilt cosine ($\tau$).** Minima I and II correspond to distinct GumK orientations. Structural snapshots highlight sampled poses, with green for the acceptor domain, blue for the donor domain, red marking GumK termini, and orange spheres indicating phospholipid phosphate groups. B) GumK crystal structure (PDB: 2HY7) from the OPM server aligned with minimum I. C) Probability density of the donor-domain distance ($d_D$) from the membrane phosphate plane, measured via Arg259 in minimum I.

## Full atomistic unbiased MD simulation

Conventional molecular dynamics (MD) simulations were performed using the CHARMM36m force field [24]. This choice was motivated by the convenient integration of CHARMM-GUI for building protein–membrane systems, as CHARMM36m provides consistent parameters for both proteins and lipids. In contrast, the AMBER99SB-ILDN force field implemented in ClustENMD is primarily tested for soluble proteins.

The crystal structure of GumK (PDB ID: 2HY7) was used for the full-atomistic simulations. The topology of the acceptor substrate was generated using the CHARMM36 carbohydrate parameters [25] and manually edited to include the undecaprenyl-diphosphate topology. The acceptor substrate was then manually positioned near the predicted binding site, based on the phospholipid interactions observed in the unbiased MD simulations.

Both the acceptor–protein complex and the free protein were embedded within a Gram-negative inner membrane using the OPM server [26]. The resulting structures were then used as input for the lipid bilayer construction module

of CHARMM-GUI to generate the membrane [27]. The lipid bilayer was composed of 1,2-dipalmitoleoyl-*sn*-glycero-3-phosphoethanolamine (DYPE), 1,2-dipalmitoleoyl-*sn*-glycero-3-phosphoglycerol (DYPG), and 1',3'-bis(1,2-dipalmitoleoyl-*sn*-glycero-3-phospho)-*sn*-glycerol (TYCL2), following a realistic phospholipid composition ratio (DYPE:DYPG:TYCL2 = 4:1:2.5) [28]. The final system was solvated using the TIP3P water model and neutralized to an ionic strength of 0.15 M.

The system underwent a multi-step equilibration process consisting of two NVT and four NPT equilibration phases to ensure proper relaxation. Position restraints were sequentially applied to the side chains, backbone, phospholipids, and trisaccharide during equilibration. Subsequently, a preliminary 10 ns production run was performed as an extension of the equilibration phase, followed by a final 1 $\mu$s production run in the NPT ensemble. The Nosé–Hoover thermostat (300 K) [29] and the Parrinello–Rahman barostat (1 bar) [30] were employed to regulate temperature and pressure, respectively.

Long-range interactions were truncated at 12 Å, and the SHAKE algorithm [31] was applied to constrain bonds involving hydrogen atoms, allowing for a 2 fs integration time step. All simulations were performed using GROMACS 2022.3 [21].

Three independent simulations of the GumK–acceptor complex were performed, each 1 $\mu$s in length, starting from the same initial structure to gather additional statistics on the system's behavior. In all unbiased MD simulations, no restraints were applied to GumK, allowing the protein to undergo conformational changes freely.

**Hamiltonian replica exchange MD simulation**

Hamiltonian replica exchange molecular dynamics (H-REMD) [32] enhances the sampling of conformational space by scaling the interactions of a selected group of atoms in the system by a factor $\lambda$. This approach reduces the number of replicas required compared to conventional replica exchange simulations, thereby lowering computational costs while maintaining efficient sampling. In the applied scaling scheme, Coulombic, van der Waals, and dihedral interaction terms were scaled, resulting in a total potential energy with a quadratic dependence on the scaling factor. This property enables the prediction of exchange probabilities between arbitrary scaling factors based on a single trial simulation using a linearly distributed set of $\lambda$ values.

For both the donor domain docked with UDP-glucuronate and UDP-glucose, scaling was applied to the sugar atoms, the pyrophosphate moiety of the substrate, and the residues in the donor domain interacting with these groups. The scaling factor ranged from 0.3 to 1.0. A total of 20 replicas were used for the UDP-glucuronate system, while 18 replicas were employed for the UDP-glucose system. The $\lambda$-ladder distribution was optimized based on an analysis of a test ladder using an in-house Python script (see GitHub).

During the simulation, two restraints were applied to reduce the complexity of the conformational space. The first was an upper-wall restraint on the DRMSD (distance root mean square deviation) of the uridine moiety docked to the protein, using as reference the optimized geometry of the crystal structure (PDB ID: 2Q6V). The second was a funnel-shaped potential applied to the $\beta$-phosphate group to prevent fully undocked conformations of the sugar.

Each replica underwent a three-step equilibration procedure. The first stage consisted of NVT equilibration at 300 K, during which the heavy atoms of the docked complex were positionally restrained. This was followed by NPT equilibration at 1 bar while maintaining the positional restraints. In the final stage, an additional NPT equilibration was performed without restraints. Throughout all equilibration steps, both the DRMSD and funnel-shaped restraint potentials remained active to prevent divergent conformations, particularly for the scaled replicas during the final equilibration phase.

Following equilibration, the replicas were propagated in the NPT ensemble at a target temperature of 300 K using the Nosé–Hoover thermostat and at a target pressure of 1 bar controlled by the Parrinello–Rahman barostat. A total of 1 $\mu$s of sampling was analyzed for both the UDP-glucuronate and UDP-glucose systems. All simulations were performed using GROMACS 2022.3 [21] in combination with PLUMED 2.8 [22].

To assess the sampling quality for the UDP-glucuronate system, we monitored the replica exchange probability, which ranged between 33% and 37%, indicating reasonable exchange between neighboring replicas. The mean square

displacement (MSD) did not fully stabilize, suggesting that some slow degrees of freedom remained insufficiently explored. The occupancy probability matrix confirmed that all replicas effectively sampled the available Hamiltonian space, although some energy states were oversampled while others remained undersampled. The replica diffusion analysis further indicated that replicas transitioned along the $\lambda$-ladder, but full diffusion across all states was not consistently achieved (S2 Fig).

For the UDP-glucose system, the exchange probability ranged between 34% and 39%, indicating frequent exchange between neighboring replicas. MSD analysis showed continuous diffusion across the replica space, confirming that replicas were not trapped. The occupancy probability matrix demonstrated that UDP-glucose explored a broader range of Hamiltonian states compared to UDP-glucuronate, with a more even distribution of replica visits. Finally, the replica diffusion analysis confirmed that UDP-glucose sampled the conformational space more effectively, with frequent transitions between states (S3 Fig).

## Bioinformatics analysis

Sequences homologous to GumI, GumH, and GumK were retrieved from UniProt [33] by searching for the respective protein names. All retrieved sequences were saved in a multi-FASTA file, and their corresponding structural models were obtained from the AlphaFold database via its API.

The donor domain of each sequence was structurally aligned with the GumK donor domain from *Xanthomonas campestris* (UniProt ID: Q8GCH2) using the FATCAT algorithm [34]. Sequences were filtered based on the quality of the alignment according to the following criteria: p-value < 0.0001, sequence identity > 8%, and exclusion of excessively short sequences. The filtered sequences were then analyzed to extract the region of each protein structurally corresponding to the sugar-binding site in GumK.

The binding site was divided into two regions, shown in purple and green in Fig 3, and saved as separate multi-FASTA files. All sequences were aligned using MAFFT [35], and a sequence logo of the resulting multiple sequence alignment was generated using WebLogo [36].

## Data availability

Data and scripts used for running and analyzing the simulations, as well as for generating the figures, are available at https://github.com/gcourtade/papers/tree/master/2025/GumK.

## Results

### The conformational space of GumK is defined by three elementary motions

GumK consists of two distinct domains, with its active site located in the interdomain region. Understanding the dynamics of these domains is crucial for identifying functionally relevant conformations (e.g., closed and open states), pinpointing key residues involved in interdomain interactions, and characterizing the complexity of its conformational space.

To achieve this, we first performed normal mode analysis (NMA) [37], a coarse-grained approach that models the protein as an elastic network in which $\alpha$-carbons act as nodes. The resulting normal modes describe intrinsic vibrational movements of the protein, with low-frequency modes capturing large-scale conformational changes and higher-frequency modes representing localized fluctuations.

The GumK crystal structure (PDB ID: 2HY7) is considered a closed conformation [12], and we used it as input for NMA. We found that the first two non-zero normal modes correspond to twisting (NM1) and bending (NM2) motions of the two domains (Fig 4A). These two modes exhibit similar frequencies, with NM1 showing a smaller eigenvalue than NM2, implying a slower collective vibration. Although the directionality of motion appears symmetric, the square displacement of the two normal modes is asymmetric (S4 Fig), likely due to the uneven distribution of interdomain contacts in the crystal structure.

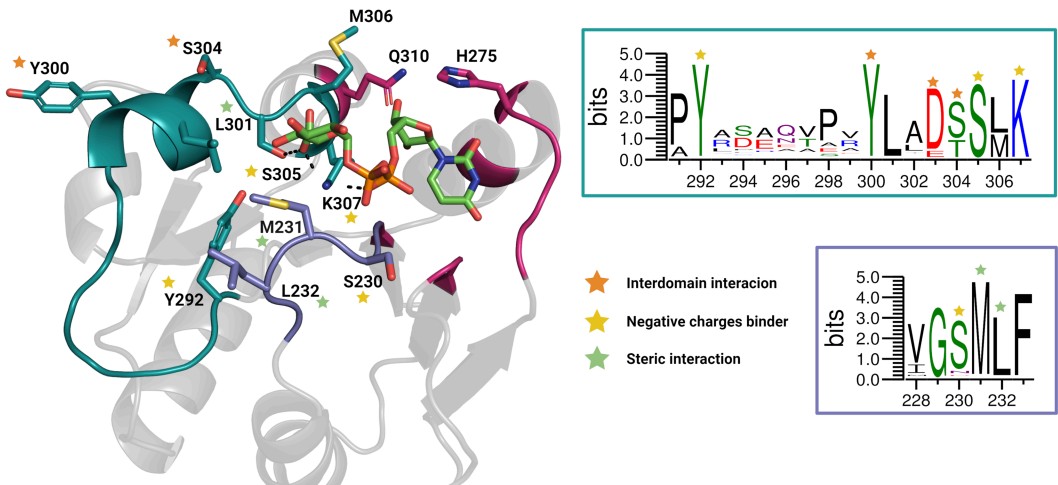

**Fig 3**. **The structure of the GumK donor-binding site is colored according to the three regions analyzed in the structural alignment.** The cyan and purple regions correspond to residues interacting with the sugar diphosphate, while the dark pink region represents the uridine-binding site. These colors match the sequence conservation analysis shown on the right. Colored stars indicate key functional residues: orange stars denote residues involved in interdomain interactions with the acceptor domain, yellow stars highlight residues interacting with both the carboxylate group and the pyrophosphate of the donor substrate, and green stars mark the hydrophobic bridge (Leu301, Met231, Leu232), which stabilizes the orientation of the sugar ring. On the right, sequence conservation is illustrated using WebLogo 3.7.12, where colors represent the chemical properties of the residues, distinguishing hydrophobic, polar, positively charged, and negatively charged amino acids.

Further analysis of the first four normal modes revealed a third key motion (NM3), characterized by a second bending of the two domains (Fig 4A). NM2 and NM3 represent perpendicular displacements: NM2 expands the interdomain cleft, increasing the distance between the two domains along their interface, while NM3 tilts one domain relative to the other, causing an asymmetric shift rather than simply widening the cleft. This third mode is the last relevant motion, as the fourth mode has a higher eigenvalue than the first three and mainly represents fluctuations localized to the acceptor domain (S4 Fig and S5 Fig).

NMA provides insight into global harmonic motions but is inherently limited to harmonic deformations around the input structure and does not account for anharmonic conformational transitions or side-chain contributions. To obtain a more comprehensive description of GumK's conformational space, we employed ClustENMD, a method that generates an ensemble of full-atomistic conformations by perturbing the structure along a subset of normal modes and subsequently refining it using the AMBER99SB-ILDN [20] force field.

The final ensemble, projected onto the first three normal modes (Fig 4B and D), reveals that as conformations shift toward more negative values of NM2, the distribution broadens, with higher RMSD values observed at the ensemble's boundary. This suggests that when contacts between the two domains are lost, the protein can adopt multiple conformations due to the flexibility of two regions spanning residues 204–221 and 353–361.

Fig 4B further shows that NM1 and NM2 are correlated, resulting in a degenerate and less informative space. In contrast, the space defined by NM2 and NM3 reveals two densely populated states: I and II (Fig 4). The first state is centered at zero, suggesting it closely resembles the crystal structure. The second state, located at negative NM2 values and positive NM3 values, corresponds to a rotated donor domain. In this conformation, helix $C\alpha4$ (Fig 1A) interacts with residues in the acceptor binding site, aligning with population II in S6B Fig.

NM3 reduces degeneracy in the conformational space as a consequence of the interdomain interface chemistry (Fig 4C). On the left side, a hydrogen bond network involving Glu192, Ser160, Tyr300, Thr161, Ser186, Asp303, and the catalytic Asp157 surrounds a hydrophobic interaction between Tyr300 and Met189. In contrast, the right side features only

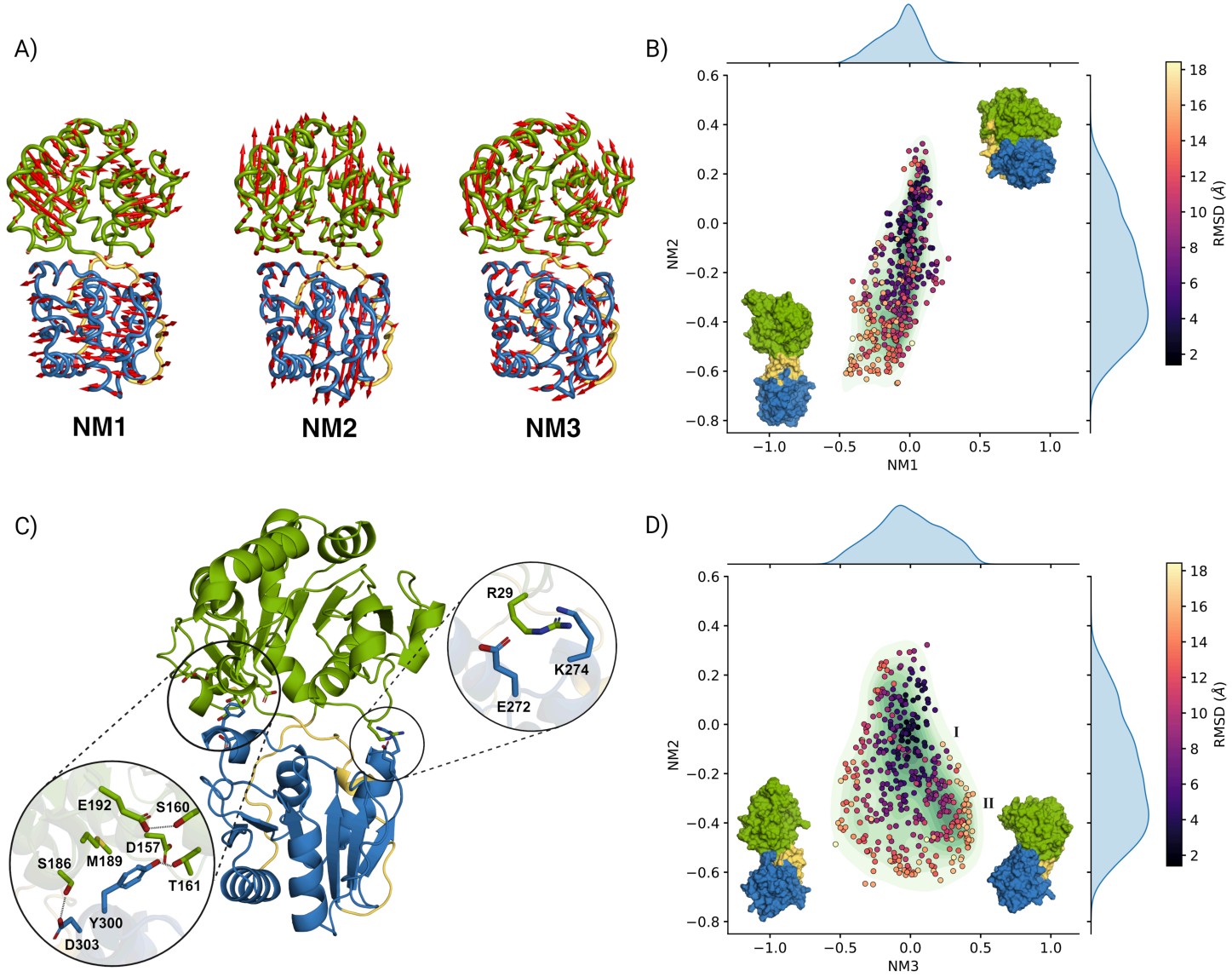

**Fig 4. A) First three normal modes from the normal mode analysis of GumK (PDB: 2HY7), highlighting its dynamic and functional properties.** The N-terminal (green), C-terminal (blue), and linker (yellow) regions exhibit distinct motions. Red arrows indicate $\alpha$-carbon displacements: mode 1 (NM1) involves symmetric twisting, mode 2 (NM2) vsymmetric bending along the main axis, and mode 3 (NM3) bending perpendicular to it. B) ClustENMD ensemble projected onto the first two normal modes. The green contour map represents population density, while the dots, colored by RMSD from the crystal structure, reveal functional motions dominated by twisting and bending. C) GumK crystal structure showing interdomain interactions: a stabilizing hydrogen-bond network and hydrophobic core (Met189, Tyr300) on the left, and a solvent-exposed salt bridge (Arg29–Asp272) on the right. These features influence domain flexibility and function. D) ClustENMD ensemble projected onto the second and third normal modes, colored as in B). Extreme conformations along NM3 show a symmetric distribution with a dense basin (population II) on the positive side.

a solvent-exposed salt bridge (Arg29–Asp272), resulting in asymmetric domain opening. These interactions are located near functional sites: the hydrogen bond network involves the catalytic residue Asp157, likely influenced by acceptor substrate binding, whereas the charged residues on the right side are close to the UDP-binding site. Thus, substrate binding could modulate interdomain dynamics and GumK activity.

The conformations sampled by ClustENMD indicate that the linkers between the two substrate-binding domains are flexible, allowing for reorientation of the domains relative to each other. To avoid confusion with established structural nomenclature, we define the acceptor domain as residues 13–203 and 362–385 (the region binding the acceptor substrate) and the donor domain as residues 222–352 (the region binding the donor substrate), with the linkers spanning residues 204–221 and 353–361 (Fig 4C).

## Charged residues in the acceptor domain mediate interactions between GumK and the membrane

GumK is hypothesized to be a monotopic membrane protein localized on the inner leaflet of the cellular membrane [38]. Based on residue properties in the acceptor domain, Barreras et al. proposed a possible membrane orientation for GumK by analyzing its crystal structure [12], but no experimental validation of this interaction has yet been reported.

To investigate how GumK interacts with the membrane, we performed MARTINI [39] coarse-grained molecular dynamics (MD) simulations using the full-length AlphaFold-predicted structure [40], which adopts a conformation similar to the crystal structure (PDB ID: 2HY7), i.e., a closed state. The predicted structure shows a backbone RMSD of 0.4 Å and a side-chain RMSD of 0.5 Å relative to the crystal structure (S7 Fig). The elastic network in MARTINI constrained the protein to the closed conformation, and OPES metadynamics [41] was employed to enhance sampling of the membrane-binding process. The closed conformation was chosen as a reference for this experiment. This choice is justified by three arguments. First, a conformation close to the experimental structure provides a reliable starting point. Second, using a closed conformation avoids sampling interdomain dynamics, which are not relevant in this experiment since the membrane interaction involves primarily the less soluble acceptor domain. Third, using the same structure for MARTINI and OPM facilitates comparison of results obtained from both methods. A cylindrical-shaped restraining potential was applied to confine the protein conformations in the unbound state and to favor the sampling of bound states (S1 Fig). This approach enabled us to model the docking process and assess the stability of different interaction modes without explicitly sampling protein conformational changes, which would otherwise make the process excessively complex.

The resulting free-energy surface (FES) (Fig 2A) is expressed as a function of the distance between the backbone bead (BB) of Leu20, approximately centered in GumK's acceptor domain, and the membrane ($d_A$), as well as its tilt angle ($\tau$) (S1 Fig). When $d_A$ is less than 5 nm, two distinct minima appear, corresponding to different orientations. The primary minimum (I in Fig 2A) has $\tau$ close to zero, with the acceptor domain interacting with the membrane through helices N$\alpha$2 and N$\alpha$4 (Fig 1A). This region is rich in arginine residues, while N$\alpha$4 contains Trp98—a key residue previously hypothesized by Barreras et al. [12] as the primary membrane-anchoring site—interacting directly with the lipid layer.

The OPM web server [26] was used as an alternative method to predict the most likely orientation of GumK relative to the lipid bilayer based on its crystal structure. The conformations within minimum I closely align with the orientation predicted by OPM (Fig 2B). The bound state is further stabilized by the N-terminus (red region in the figure), which can also interact with the membrane. Analysis of the donor domain positioning in this state reveals two possible conformations (Fig 2C). In one, the donor domain interacts with the phospholipid layer, slightly increasing $\tau$; in the other, the donor domain remains free in solution. The second minimum (II in Fig 2A) of the FES corresponds to an opposite orientation, where both termini of the acceptor domain interact with the membrane.

Overall, the OPES simulations sampled the expected binding mode, consistent with the hypothesis of Barreras et al. [12] and supported by the OPM prediction. The second minimum is not compatible with the typical orientation of membrane-associated GT-B enzymes [17], but it suggests that both termini may transiently associate with the membrane, potentially contributing to further stabilization of the protein–membrane complex.

The simulation was not fully converged, as the bias potential was still being deposited at the end of the run in the region where the protein binds to the membrane (as predicted by OPM). This indicates that, although this binding mode was sampled, the corresponding minimum was not fully explored, and the resulting free-energy surface should therefore be interpreted qualitatively.

Although ClustENMD effectively explored conformational changes (Fig 4), it lacks membrane context, limiting its ability to fully capture GumK's behavior. Additionally, understanding the kinetics of the opening mechanism and how the protein behaves without a constraining potential bias is crucial. For these reasons, we performed fully atomistic MD simulations using the OPM-derived orientation in a solvated membrane system.

The system was simulated for 1 $\mu$s, and conformational changes were analyzed using essential dynamics analysis (EDA) [42], employing principal component analysis (PCA) to construct a covariance matrix based on $\alpha$-carbon displacements throughout the trajectory. As with normal mode analysis, the resulting principal components (PCs) describe the protein's conformational changes sampled during the simulation. A total of 20 PCs were computed using ProDy [42]. The eigenvalue plot (S8 Fig) indicates that the first six PCs capture the majority of the variance, with PC1 alone explaining 67%. The non-Gaussian distributions of these components suggest that they define GumK's essential subspace (S9 Fig) [43]. PC1 exhibits a bimodal distribution, with one peak near zero corresponding to a conformation similar to the crystal structure (i.e., closed), while a second peak indicates a more open state. The cosine content of the time series for this PC is 43.1%, indicating that the transition leading to the bimodal distribution corresponds to a conformational change rather than the exploration of a broad basin [44].

Projection of PC1 onto the first three normal modes (Table 2) reveals that it is a linear combination of NM2 and NM3, representing an opening motion between domains. This motion aligns with the third quadrant of the NM3 vs. NM2 plot. An additional ClustENMD simulation, sampling conformations closer to the crystal structure, confirmed a strong correlation between NM2 and NM3, reinforcing the interpretation that PC1 captures the protein's opening transition (S10 Fig).

PC2 follows a more Gaussian distribution, though it retains a high eigenvalue ( S8 Fig and S9 Fig). Projection onto the normal mode space (Table 2) indicates that PC2 combines NM1 and NM3, corresponding to a twisting motion of the two domains.

To assess the convergence of the simulation, we used the bootstrap normalized block covariance overlap method. Fitting the inverse of the resulting curve with a triple exponential provides an estimate of the time required for convergence. The analysis shows that the system is far from converged, and at least 4 $\mu$s would be required to achieve full decorrelation of the covariance overlap matrix (S11 Fig) [45]. However, the two principal component directions converge more rapidly, as shown in S12 Fig. Since we are primarily interested in describing the opening pathway of the protein and defining a space that discriminates between different conformations, the convergence of the PC directions is more relevant.

By projecting the MD trajectory onto the PC1–PC2 space (Fig 5A), two main populations were identified: one corresponding to the crystal-like closed state and the other to a more open conformation.

The full-atomistic simulations confirmed the observations made in the MARTINI coarse-grained model, showing that over 1 $\mu$s, the donor domain can also interact with the membrane. Evidence for this interaction is found in the distribution of distances between Arg259 and the phospholipid headgroup plane (Fig 5D), which closely resembles that observed in the MARTINI simulations (Fig 2C). Moreover, additional residues such as Glu236 and Asp234 interact with the membrane via electrostatic attraction to the positively charged phosphatidylethanolamine headgroups.

Further analysis of the acceptor domain revealed key hydrophobic and electrostatic interactions stabilizing GumK at the membrane. Trp98 and Trp109, located on helix N$\alpha$4, are inserted into the membrane, anchoring the protein (Fig 1A).

During the first 400 ns, a cardiolipin (TYCL2) molecule binds within the cleft between helices N$\alpha$4 and N$\alpha$2 (Figs 1A and 5C), while a phosphatidylglycerol (DYPG) molecule binds to a groove between helices N$\alpha$2 and N$\alpha$3.

**Table 2**. **Projections of the first two principal components computed in the essential dynamics analysis of the GumK-membrane complex unbiased MD simulation.**

| Vector | NM1 | NM2 | NM3 |
|---|---|---|---|
| PC1 | +0.09 | +0.43 | +0.12 |
| PC2 | −0.13 | −0.02 | +0.34 |

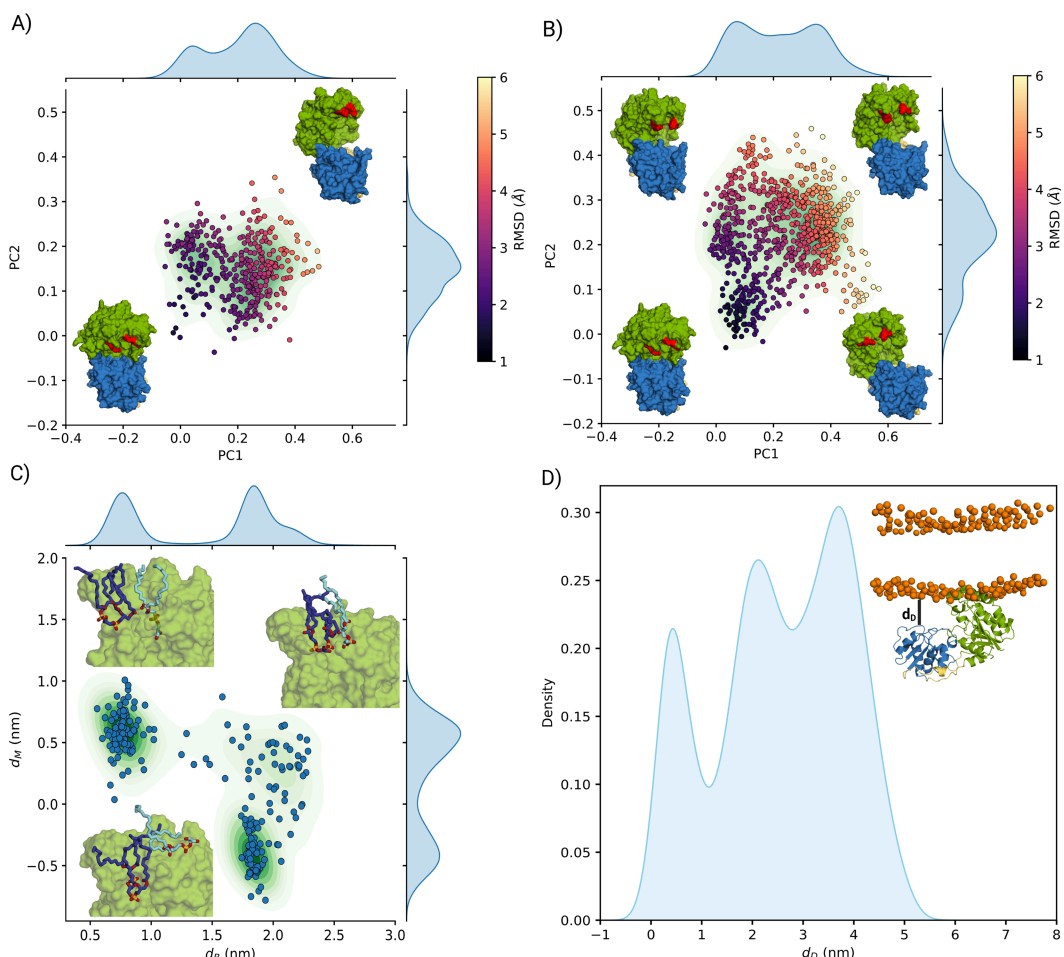

**Fig 5**. **A) Projection of the conformations sampled during the MD simulation onto the first two principal components of the essential subspace.** The green contour map represents the population density of the total ensemble, while individual conformations are shown as colored dots based on their RMSD relative to the GumK crystal structure. The two structures highlight representative conformations: a closed conformation on the negative side of the axes and a partially open conformation on the positive side. The red region marks the acceptor-substrate binding site of the tail, used as a visual guide. B) GumK conformations sampled in the presence of the acceptor substrate, projected onto the same space as panel A). The representative structures illustrate how the protein explores the four quadrants: open in the first, twisted-closed in the second, closed in the third, and twisted-open in the fourth. C) Binding path of phosphatidylglycerol (DYPG) sampled during the simulation. The x-axis represents the distance of the DYPG phosphorus atom from the center of mass of the binding region ($d_B$), while the y-axis shows the distance of the same atom from the average plane of the membrane phosphate heads ($d_M$). The displayed conformations are representative of the binding path. D) Distance distribution of the donor domain from the membrane ($d_D$) as shown in the structure. The $\alpha$-carbon of residue Arg259 is used as a reference for this distance, indicating that the donor domain interacts with the membrane.

After 400 ns, DYPG gradually moves from the top of the acceptor domain toward the bound TYCL2, eventually replacing it. During this process, DYPG is pulled out of the membrane and interacts with Lys60, His90, Arg52, Ser57, and Arg64, located in the N$\alpha$4–N$\alpha$2 cleft (Figs 1A and 5C), while the departing TYCL2 interacts with Arg95 and Asn93. The lipid tails of both phospholipids interact with hydrophobic residues in the cleft, particularly Leu56, Val89, and Phe92, which form a groove on the acceptor domain capable of accommodating the hydrophobic tails. The phospholipid binding events observed in these simulations suggest the likely location of the acceptor substrate–binding site and provide insights into its orientation within the cleft.

## GumK donor binding site discriminates between its GlcA-UDP substrate and the non-reactive ligand Glc-UDP

Although Barreras *et al.* attempted to co-crystallize an inactive mutant of GumK with its donor substrate, GlcA-UDP [12], the available X-ray crystal structure (PDB ID: 2Q6V) contains coordinates only for the UDP moiety. This means that the binding pose of the donor substrate's sugar portion remains unknown. To address this, we performed fully atomistic simulations using the CHARMM36 force field [24] and the enhanced-sampling Hamiltonian Replica Exchange (HREX) [46] method to explore the conformational space of the monosaccharide.

To facilitate the exploration of this space, we adopted a three-step approach. First, we simplified the system by simulating only the isolated donor domain instead of the full-length protein to avoid bias from overall protein conformation. Second, we employed HREX to enhance sampling efficiency. Third, we constrained the uridine moiety to its crystallographically determined binding conformation, while the sugar portion was confined using a funnel-shaped restraint to prevent unbound conformations. Three independent HREX simulations were performed, with 19 and 17 replicas, respectively, for GlcA-UDP and Glc-UDP. Results from the first simulation are shown in S2 Fig, while the remaining two were used to assess reproducibility (S13 Fig).

For GlcA-UDP, the final replica exchange probability ranged between 33% and 37%, indicating reasonable exchange between neighboring replicas. The replicas moved freely along the replica ladder, and no serious clustering was observed; however, full diffusion was not achieved (S2 Fig). We used the distance ($d_k$) between the C6 atom of GlcA-UDP and the $N^\zeta$ atom of Lys307 as a collective variable (CV) to characterize the binding modes of the sugar moiety. The populations sampled during the simulation are shown in S2C Fig. The distance distribution revealed two predominant binding modes for GlcA-UDP: one centered at $d_k$ = 0.3 nm and another at $d_k$ = 0.9 nm. In the first mode, the carboxylate group of GlcA and one of the two phosphate groups interact with Lys307. In the second mode, only the phosphate groups interact with Lys307, while the carboxylate group remains solvent-exposed.

Although $d_k$ provides valuable insight into substrate binding, it does not distinguish between different orientations of the sugar ring. To achieve a more detailed characterization of the binding modes, we introduced two additional angular CVs, $\gamma$ and $\theta$, defined as the angles between the sugar ring's normal vector and the donor domain's principal axes of inertia, describing the orientation of the sugar ring relative to the protein (S2D Fig).

Analysis of the polar coordinate space revealed three well-defined populations (Fig 6). Cluster I is centered at $d_k$ = 0.9 nm and features the sugar ring oriented with its $\beta$-face toward helix C$\alpha$7. Cluster II exhibits an average $d_k$ = 1.02 nm, maintaining the $\alpha$-face orientation toward C$\alpha$7 but in a distinct spatial arrangement. Cluster III represents a broad and flexible basin, in which the sugar ring is more solvent-exposed while both phosphate groups interact with Lys307, allowing greater conformational variability. A fourth cluster, Cluster C, has an average $d_k$ = 0.35 nm, with the $\alpha$-face directed toward C$\alpha$7.

To evaluate the convergence of these basins, the RMSD of the free-energy surface (FES) as a function of the three CVs ($\gamma$, $\theta$, and $d_k$) was plotted over an increasing number of frames. The reference FES corresponds to the starting point of the simulation (S14A,B Fig). All FESs were computed by combining all replicas and using the binless WHAM algorithm to estimate the weights for the weighted histogram. These analyses show that although the replicas did not achieve full diffusion along the replica ladder, the FESs as functions of the target CVs appear to converge.

We then analyzed the sugar-binding region to identify key interactions contributing to substrate recognition and stabilization. Tyr292, Ser305, and Lys307 interact with both the carboxylate and phosphate groups across different binding clusters. Additionally, Leu301, Met231, and Leu232 form hydrophobic interactions with one another, effectively excluding the flexible loop between Ala293 and Pro298 from direct contact with the sugar moiety. The distances among these hydrophobic residues vary across clusters (S15 Fig), suggesting that they adopt distinct configurations depending on the GlcA-UDP binding mode, potentially influencing the orientation of the sugar ring.

It has been hypothesized that GumK exhibits high specificity for its native substrate, GlcA-UDP [47]. To further investigate this selectivity, we performed an additional HREX simulation using the same setup but with UDP-glucose (Glc-UDP),

PLOS Computational Biology

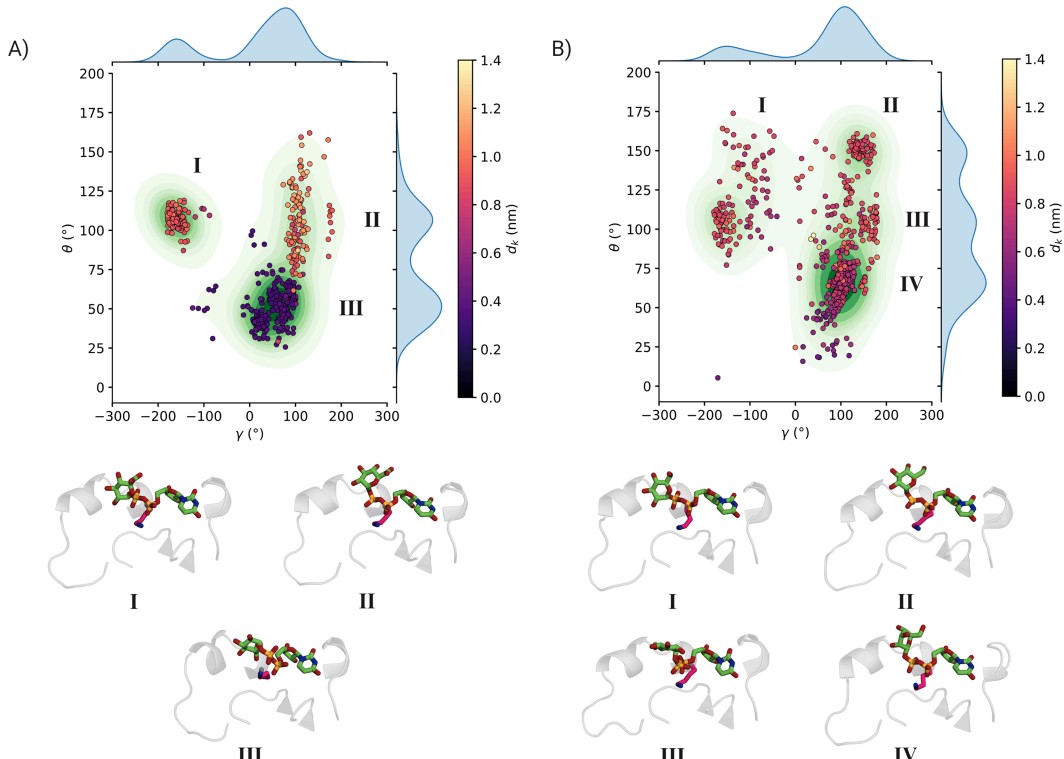

**Fig 6**. **A) Binding modes of UDP–glucuronate sampled during a constrained Hamiltonian replica-exchange simulation of the donor domain.** The two axes represent key angles defining the sugar-ring orientation relative to the protein (S2D Fig). The color scale indicates the distance between the sugar C6 atom and the $N^\zeta$ atom of residue Lys307 (shown as a pink stick in the structures), while the green contour map represents the population density of the sampled conformations. Three distinct clusters, differing in both distance and orientation, are identified. Representative structures for each cluster are shown below. B) Binding modes of UDP–glucose sampled using the same simulation protocol and plotted in the same space as panel (A) (S3D Fig). In this case, four clusters are observed, with representative structures displayed below.

which is not a GumK substrate [18]. For this substrate, the exchange probability ranged between 34% and 39%, indicating frequent exchange between neighboring replicas. The replicas spanned the replica ladder, but as in the GlcA-UDP case, full diffusion was not observed (S3 Fig). The distance distribution for Glc-UDP revealed a single dominant population centered at $d_k$ = 0.8 nm (S3C Fig), corresponding to binding modes in which the phosphate groups interact with Lys307. Unlike GlcA-UDP, glucose's C6 does not interact with Lys307, Tyr292, or Ser305. Analysis in polar space (Fig 6B and S3D Fig) indicated greater conformational flexibility for Glc-UDP, highlighted by an additional peak (Cluster II), where the sugar ring adopts a perpendicular orientation relative to helix C$\alpha$7. On average, all visible clusters exhibit $d_k$ values between 0.7 and 0.8 nm. Also in this case, the hydrophobic interactions in the binding site exhibit a degree of flexibility that depends on the orientation of the sugar motif (S16 Fig). The same convergence analysis performed for Glc-UDP yielded similar behavior, with the RMSD tending to flatten over time (S14C,D Fig).

HREX effectively explored the sugar–diphosphate binding modes when the nucleoside portion of the donor substrate was constrained. The donor domain structure appears to impose conformational constraints on the substrate, with the acidic substrate being more restricted than the neutral one due to the additional interaction between the carboxylate group and Lys307. Although free diffusion along the replica ladder was not achieved, the sampled basins were reproducible (S13 Fig), indicating that the qualitative behavior of the two substrates is consistent. However, the

lack of free diffusion might indicate incomplete sampling of the polar coordinate subspace used to describe sugar conformations.

## Bioinformatic analysis shows conserved residues in the donor domain sugar-binding site

The previous simulations identified key residues in the sugar-binding site—Tyr292, Ser305, Lys307, Met231, Leu232, and Leu301—strategically positioned to orient the sugar ring. If these residues play a functionally critical role in the donor domain, their spatial arrangement would be expected to be conserved across different GumK enzymes. To assess this hypothesis, we performed a structural alignment of all sequences labeled as GumK in UniProt, comparing each protein's predicted structure with the crystal structure of GumK.

Similarly, we conducted structural alignments for GumI and GumH (S17 Fig), two enzymes involved in xanthan gum side-chain biosynthesis that act on mannose-GDP (Man-GDP). GumI belongs to the CAZy GT94 family, while GumH is classified within the GT4 family. For these enzymes, we used the AlphaFold-predicted models of *Xanthomonas campestris* GumI and GumH as structural references.

One of the most distinctive features of GumK is the presence of a conserved hydrophobic interaction network formed by Leu301, Met231, and Leu232, which plays a critical role in constraining the orientation of the sugar ring (Fig 3). This interaction is closely associated with the loop spanning Ala293 to Pro298, which, as shown by simulations, contributes to the flexibility of the binding region. While this loop is structurally conserved among GumKs, its amino acid composition is highly variable and does not exhibit a conserved chemical nature. The absence of this loop in GumI and GumH suggests that it represents a structural element specific to GumK, potentially contributing to the enzyme's specificity and flexibility in sugar recognition.

Another key difference lies in the distribution of positively charged residues in the binding site. GumK is characterized by the presence of a conserved lysine (Lys307) at the donor-binding site, which interacts with the carboxylate group of GlcA-UDP. Structural alignment revealed that while GumKs consistently maintain a lysine at this position, GumI and GumH typically feature a neutral residue instead. However, both GumI and GumH display two positively charged residues—a Lys–Arg pair (S17 Fig)—located in a different region of the phosphorylated sugar-binding site, indicating an alternative charge-distribution strategy for stabilizing the binding of Man-GDP. Furthermore, Tyr300 and Ser304, which contribute to the hydrogen-bond network stabilizing interdomain interactions (see Fig 4C), are structurally conserved across all GumKs.

Based on these observations, the donor monosaccharide-binding region can be divided into four functional segments (Fig 3): (i) the region formed by helices C$\alpha$4 and C$\alpha$7 (Fig 1A), including residues Tyr300 and Ser230, which interact with the acceptor domain and orient the sugar ring while engaging the acidic group of GlcA-UDP; (ii) the flexible loop between Ala293 and Pro298, unique to GumK—conserved in topology but variable in sequence—and absent in GumI (GT94) and GumH (GT4); (iii) the hydrophobic interaction triad formed by Leu301, Met231, and Leu232, a feature exclusive to GumK that stabilizes sugar-ring orientation; and (iv) the uridine-binding region.

## The acceptor substrate is constrained in a linear conformation by the binding site

GumK specifically binds to the acceptor substrate Man-Cel-UNDPP (Fig 1B), showing a clear preference for interactions with the pyrophosphate group [47]. The hydroxyl group at the C2 position of the mannose unit serves as the reactive site, forming a hydrogen bond with the catalytic residue Asp157.

To explore the molecular basis of GumK–acceptor interactions, we performed three molecular dynamics (MD) simulations of the GumK–acceptor complex in a membrane environment. The initial complex was constructed by manually positioning the acceptor's tail near the region that interacted with DYPG in the free-protein simulation (Fig 5C and S18 Fig). Each replica was simulated for 1 $\mu$s, resulting in a total cumulative time of 3 $\mu$s. The distance between the acceptor's

reactive oxygen and the carboxylate carbon of Asp157 ($d_{acc}$) was used as a collective variable (CV) to monitor the binding event.

Among the three replicas, only one maintained the acceptor substrate in close proximity to the catalytic residue, adopting a reactive conformation for an extended period. Therefore, only this trajectory was considered for further analysis (Fig 7A and S19 Fig).

During the first 200 ns (region I in Fig 7A), the acceptor transiently interacts with Arg29, disrupting the interdomain salt bridge (Arg29–Glu272; Fig 4C) that maintains the protein in a closed conformation. Despite the flexibility of the trisaccharide, the substrate remains near the binding site due to the stable anchoring of its lipidic tail within the cleft between helices N$\alpha$2 and N$\alpha$4 (Fig 1A).

After 200 ns, the acceptor adopts a more constrained conformation, with $d_{acc}$ remaining below 1 nm for approximately 600 ns (region II in Fig 7A). During this period, the trisaccharide adopts a predominantly linear conformation, as reflected by a narrower dihedral distribution for the $\alpha(1{\to}4)$ glycosidic bond compared to the unbound state. Although the $\alpha(1{\to}3)$ bond exhibits a broader dihedral distribution, it still favors a linear conformation (S20 Fig). Within this time window, $d_{acc}$ alternates between two states: one in which the reactive hydroxyl of mannose forms a hydrogen bond with Asp157, and

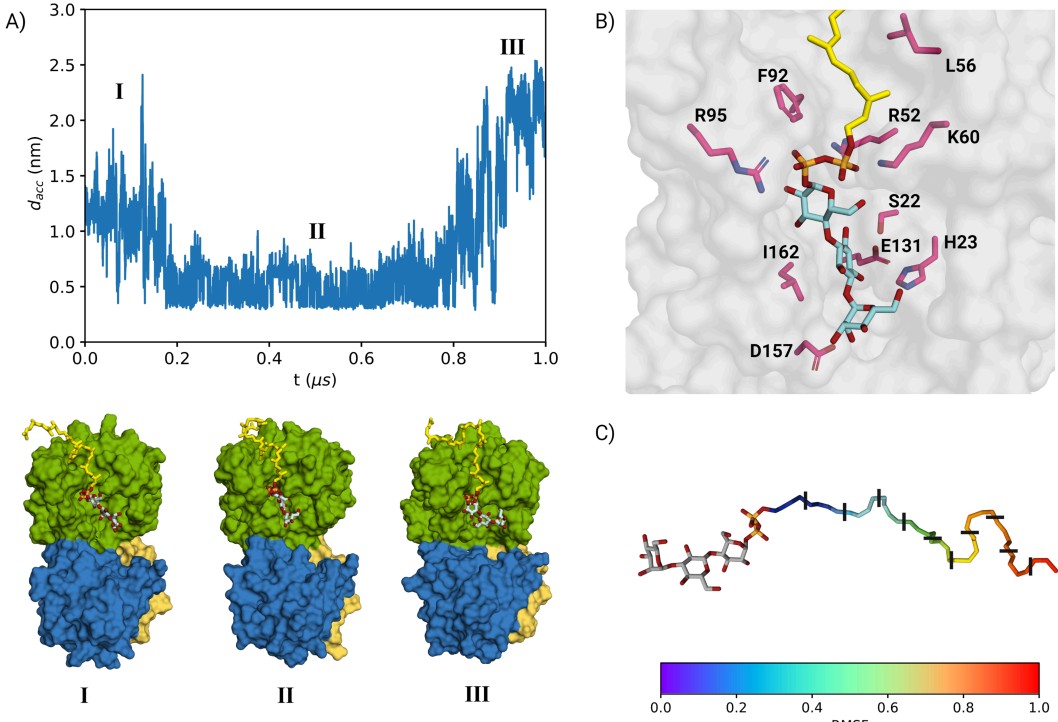

**Fig 7. A) Time evolution of the distance between the acceptor substrate and the binding site, measured between the reactive oxygen at C1 of the last sugar unit and the carboxylate carbon of Asp157.** The system transitions through three main states, illustrated by the structures at the bottom: I, II, and III. The bound state (II) corresponds to a more open protein conformation, in which the acceptor substrate is stabilized within the binding site. States I and III represent unbound conformations, where the substrate interacts with Arg29, promoting protein opening. B) Detailed representation of the bound state, highlighting key residues involved in substrate recognition and binding. C) Normalized RMSF (root mean square fluctuation) of the carbon atoms in the lipidic tail of the acceptor substrate in the bound state. Black lines mark the isoprenyl units, showing that the first three units remain stably bound within the binding pocket.

another in which it remains near the catalytic residue without direct contact. This behavior indicates that the acceptor substrate retains a degree of conformational freedom, likely due to the absence of the donor substrate, which would otherwise impose additional structural constraints. Furthermore, the protein undergoes partial opening upon acceptor binding, increasing the conformational flexibility of the bound substrate.

Additional contacts help stabilize the substrate, including interactions between the C6 atom of $\beta$-glucose and Ser22, C4 and Ser132, and $\alpha$-glucose with Arg52 and Ser22. Furthermore, the C6 atom of $\alpha$-mannose interacts with His23, whose orientation is stabilized by Glu131 and Ser22. A stable hydrogen-bond network involving Glu131, Ser22, and Ser132 maintains these residues properly oriented within the binding site (Fig 7B). However, the bound state remains dynamic, characterized by short-lived contacts that suggest the presence of multiple conformations rather than a single stable binding pose.

The lipid tail follows a distinct binding pattern: the first three isoprenyl units remain stably bound, whereas the rest of the tail is more flexible, as indicated by the RMSF of the tail carbons (Fig 7C). This stability originates from the cleft between helices N$\alpha$2 and N$\alpha$4 (Fig 1A), which encloses the tail and enables hydrophobic interactions. The second isoprenyl unit engages Phe92 and Leu56, contributing to substrate retention. Notably, the unsaturated bond of this unit stacks against the aromatic ring of Phe92, further stabilizing the substrate within the binding pocket.

To evaluate how acceptor binding affects GumK dynamics, we projected the trajectory onto the essential subspace previously defined for GumK (Fig 5B). The analysis revealed that acceptor binding promotes protein opening, with the domains separating more rapidly and remaining open longer than in the unbound state. This effect is primarily driven by interactions with Arg29, which destabilize the interdomain salt bridge and facilitate domain separation.

Additionally, the acceptor disrupts the interdomain hydrogen-bond network, particularly the Ser160–Glu192 interaction, increasing domain twisting and further promoting opening. During the simulation, residues within this network remain in proximity but undergo transient rearrangements. In particular, a new hydrogen bond between Tyr328 and His201 helps maintain domain proximity, while Tyr300 shifts beneath the N7a turn (Fig 1A).

The pyrophosphate group of the lipid tail serves as the main driver of acceptor binding, clamping tightly between helices N$\alpha$4 and N$\alpha$2 through a combination of electrostatic and hydrophobic interactions. This interaction stabilizes the substrate, keeping the trisaccharide close to the binding site and inducing conformational changes at the interdomain interface. However, as the reactive hydroxyl approaches Asp157, the binding becomes less stable, suggesting that lipid-tail anchoring plays a dominant role in the initial substrate recognition and retention process.

## Discussion

GT-B enzymes exhibit remarkable conformational plasticity, with diverse domain orientations driven by distinctive twisting and bending motions. Our multiscale computational investigation of GumK reveals the molecular mechanisms underlying this flexibility. This structural dynamism plays a fundamental role in substrate recognition and catalysis, establishing a direct connection between substrate binding, conformational transitions, and enzymatic activity.

### The interdomain contacts can tune GumK activity

Normal mode analysis highlighted three key motions: a twisting mode (NM1) and two bending modes (NM2 and NM3) (Fig 4A). However, the topology of the elastic network depends on the initial conformation, as inter-residue contacts are defined based on a radial cutoff. In the crystal structure (PDB ID: 2HY7), the hydrogen-bond network forms a more compact interface than the salt-bridge side (Fig 4C). As a result, NM1 is the slowest mode, as it involves disruption of interdomain contacts. Conversely, analyzing open GumK conformations yields different results due to the reduced number of interdomain interactions. Starting from a closed conformation—such as the one observed in the crystal structure—provides a more meaningful set of normal mode vectors, as this state preserves the potential contacts between domains. In contrast, when the protein adopts an open conformation, the eigenvalues of the first three modes approach zero (S5B

Fig), indicating that these modes correspond primarily to nearly free rotations and translations of the two domains. This occurs because, in the absence of interdomain contacts, the protein behaves approximately as two independent domains that move freely relative to one another, as also shown by the ClustENMD results (Figs 4B and C).

ClustENMD provided additional insights into GumK's domain dynamics at the atomistic level. The NM2–NM3 projection (Fig 4) revealed that bending motions broaden the conformational space due to the flexibility of the linker regions (residues 204–221 and 352–361). The bending pathway is strongly influenced by the initial interdomain interaction pattern, as demonstrated by the small perturbation of GumK in ClustENMD (S10 Fig). When these interactions are completely lost, the two domains become fully independent, leading to an expansion of the conformational space. This increased flexibility makes it less likely for the protein to recover its starting (closed) conformation, which, in functional terms, translates into a reduction in catalytic efficiency (i.e., a decrease in $k_{cat}$).

This effect is particularly evident in the sampling of non-active conformations, such as the overtwisted state (S6A Fig) and state II (Fig 4D). The overtwisted state is incompatible with enzymatic activity, as the donor substrate–binding site is positioned behind the acceptor substrate–binding site. In state II, helix C$\alpha$7 occupies the acceptor-binding pocket, preventing substrate interaction.

It is important to note that ClustENMD may generate unphysical conformations characterized by distorted secondary structures and/or overstretched loops, due to the external forces applied along the deformation vectors [48]. Not all structures sampled by ClustENMD necessarily belong to the ensemble of conformations populated in solution. The applied force tends to deform the secondary structure of the domains rather than promoting a genuine opening, leading to unphysical distortions. To quantify the extent of these deformations, we compared each domain to the starting structure using RMSD. In the acceptor domain, 8.9% of the conformations exceed an RMSD of 2 Å, while in the donor domain this fraction is 3.1%. Furthermore, a subset of the conformations sampled by ClustENMD is also observed in the fully atomistic MD simulations (S10 Fig), supporting the reliability of ClustENMD as a method to generate open conformations of GT-B glycosyltransferases.

The stable hydrogen-bond network (Fig 5C) strongly influences the conformational landscape. Experimentally disrupting these interactions—for instance, by mutating key residues such as Tyr300—could weaken interdomain contacts, enhance twisting motions, and favor open conformations. This shift might ultimately reduce catalytic efficiency by promoting inactive states.

## The membrane affects the interdomain interaction by keeping the domains closer

In a cellular environment, domain mobility is further constrained by intracellular crowding, which favors compact conformations [49], and by membrane interactions that contribute to the stabilization of GumK. While future advancements may enable fully realistic simulations of cellular environments, current computational approaches remain limited to modeling proteins anchored to membranes.

Both the MARTINI-OPES simulations and the OPM server analysis indicated that the protein interacts with the membrane through the amphipathic helix N$\alpha$4. This anchoring is further stabilized by two tryptophan residues located on this helix, which have been reported to act as characteristic anchoring points in monotopic proteins. These findings suggest that GumK is membrane-associated and not in equilibrium with the soluble fraction, consistent with previous reports [12, 38].

Furthermore, the orientation of the acceptor domain constrains the overall orientation of the protein, leaving little room for large conformational flexibility at the membrane, apart from minor fluctuations. This constraint arises from the structural properties of the protein itself and not from simulation restraints. Any substantial change in the tilt angle would decrease the contact between the tryptophan residues and the membrane, leading to a less stable anchoring. Therefore, we expect that the binding mode would not significantly differ even if the OPES simulations were performed starting from the open conformation.

To test this, we performed an OPM prediction on one of the open conformations sampled by ClustENMD, and the result showed that the acceptor domain is oriented in the same way (S21 Fig). However, this comparison may not hold for a quantitative free-energy analysis, since the solvation sphere of the open and closed conformation differs. Nonetheless, the main goal of this work was to characterize the binding mode of the protein at the membrane rather than to directly compute a binding free energy.

Both MARTINI coarse-grained and unbiased all-atom MD simulations showed not only that the acceptor domain binds to the membrane as expected, but also that the donor domain can interact with the bilayer through electrostatic contacts. This additional interaction has previously been described for other membrane-associated GTs [17]. Plant GT-B enzymes, such as atDGD2, represent an extreme example, as both domains engage in membrane anchoring via tryptophan residues [50]. Furthermore, membrane-dependent enzymatic activity has been reported for PimA, a GT4 glycosyltransferase involved in phosphatidylinositol mannoside biosynthesis in *Mycobacteria*. Despite being a peripheral enzyme, PimA exhibits increased activity in the presence of acceptor substrates or lipid molecules above the critical micelle concentration [51]. In GumK, donor-domain interactions with the membrane may affect interdomain dynamics by keeping the domains in closer proximity. This behavior suggests that the lipidic environment may influence GumK by promoting compact and functionally relevant conformations.

In the MARTINI OPES simulation, the protein was constrained to a closed conformation, meaning that variations in donor-domain distance primarily result from minor acceptor-domain tilts within the same energy minimum, influenced by membrane interactions. These findings, along with observations of interactions between the two protein termini (Fig 2A), suggest that the protein can adopt different binding modes and may adapt to distinct membrane topologies. The simulation indicates that both termini can interact with the membrane. Since AlphaFold assigns a low confidence score to this region, the structural arrangement of the termini remains uncertain. Although they were modeled as random coils in the MARTINI simulation, it is possible that they adopt an $\alpha$-helical fold that enables stable membrane binding. The specific orientation of GumK relative to the membrane is strongly influenced by the topology of the interfacial residues, particularly the presence of Trp98, which acts as a key anchoring point. This orientation may vary among different GT-Bs, likely reflecting the intrinsic dynamics of the protein backbone.

During 1 $\mu$s of unbiased MD simulation, GumK exhibited multiple conformational changes (Fig 5), demonstrating its ability to adopt open states even in the absence of substrates. While ClustENMD provided a qualitative representation of the conformational landscape, quantitative analysis requires enhanced-sampling methods. Protein opening in this fold is often modeled using two collective variables (CVs): an interdomain angle describing bending and a dihedral angle describing twisting [52]. However, metadynamics along these CVs requires extensive restraints to avoid irrelevant phase-space sampling [53]. ClustENMD conformations suggest a similar challenge for GumK. Nevertheless, essential dynamics analysis identified PC1 and PC2 as potential pathways describing domain opening and twisting (Figs 5A, B; Table 2), which could serve as more effective CVs for future enhanced-sampling simulations.

## The acceptor substrate is clamped by GumK and enhances protein opening

GumK fluctuations in the simulations keep the donor-binding site partially buried (Fig 5A), suggesting that spontaneous opening may not be sufficient for donor binding. Conversely, the acceptor-binding region remains accessible, as shown by the interactions between the acceptor domain and phospholipids (Fig 5C), which are clamped within the cleft between helices N$\alpha$2 and N$\alpha$4. Additionally, several annular phospholipids at the protein–membrane interface exchange slowly with the membrane bulk due to electrostatic interactions with arginine residues at the membrane-binding region. Notably, a phosphatidylglycerol molecule involved in phosphatidylcholine replacement, observed during the simulation (Fig 7C), remains bound for a long time in a groove between helices N$\alpha$2 and N$\alpha$3. This region may play a key role in the acceptor-substrate binding pathway, facilitating substrate recognition and interaction with the active site.

When the acceptor is positioned similarly to the bound phosphatidylglycerol, the protein still tends to clamp the poly-isoprenyl tail, enclosing the pyrophosphate region. Experimental studies by Barreras *et al.* showed that GumK remains active on shorter phytanyl tails containing only four isoprenyl units, suggesting that substrate specificity is less dependent on tail length [47]. Our simulations align with this observation, showing that while the first three isoprenyl units remain stably bound, the remainder of the molecule exhibits greater flexibility (Fig 7C). Additionally, Phe92 interacts with the first isoprenyl unit via $\pi$–stacking, reinforcing the importance of unsaturated lipid chains in substrate binding. These findings indicate that GumK can accommodate shorter oligo-isoprenyl substrates, provided they retain unsaturation, in a manner reminiscent of the substrate tolerance observed in the GT1 glycosyltransferase MurG from *E. coli* [54].

The trisaccharide portion of the acceptor substrate promotes enzyme opening by interacting with the salt bridge between Arg29 and Glu272 (Fig 7A). In contrast, phospholipids bound to the N$\alpha$2–N$\alpha$4 cleft (Fig 5C) do not induce significant opening, as they are too short to reach the interdomain space. These observations suggest a degree of independence between the domains: while phospholipid binding induces local rearrangements at the N$\alpha$2–N$\alpha$4 cleft, only the acceptor substrate, which interacts at the interdomain interface, triggers full opening.

Based on these results, we propose that acceptor binding represents the initial step in GumK opening, creating space for subsequent donor binding. This implies that state II identified in ClustENMD (Fig 4D)—which becomes more probable when interdomain contacts are lost—is disfavored in the presence of the acceptor substrate, as it occupies the acceptor cleft and prevents donor-domain interactions (S6B Fig). Consequently, the presence of the acceptor substrate would reshape the conformational landscape shown in Fig 4D by reducing population II.

This mechanism aligns with findings in GT4 glycosyltransferases, particularly PimA, which transfers mannose from Man–GDP to phosphatidyl-*myo*-inositol in *Mycobacterium* [55]. In PimA, the acceptor substrate induces a relaxed conformation, whereas donor binding stabilizes the enzyme in a closed state [56]. Similar substrate-dependent conformational transitions are observed in other members of the GT4 CAZy family [57,58].

During the unbiased simulations, only one state—lasting approximately 600 ns over the total 3 $\mu$s—can be considered close to a bound state. The substrate remains near the catalytic residues through hydrogen bonds with the backbone, stacking interactions with Arg52 and Ile162, and electrostatic/hydrophobic contacts between the lipid tail and the amphiphilic acceptor-domain clamp. The binding site constrains the sugar moiety in an extended conformation, suggesting that volume, regiochemistry, and stereochemistry all influence substrate specificity. The lipid tail, analogous to the UDP motif in the donor substrate, appears to guide and anchor binding.

These structural features suggest that GumK may tolerate various trisaccharides, provided they fit within the binding site. This interpretation is consistent with experimental data showing reduced activity on acetylated acceptor derivatives [47].

## The selectivity for the donor substrate is partially affected by the structure of the donor's binding site

It has been reported that GumK is highly selective for GlcA–UDP [47]; however, direct experimental evidence of its donor interaction comes from two crystal structures (PDB: 2Q6V and 3CV3), where only the UDP moiety is visible due to spontaneous hydrolysis of GlcA–UDP. Structural data show that the UDP $\beta$-phosphate interacts with Lys307 and Tyr292, and mutating these residues to Ala abolishes catalytic function [12]. However, the presence of the GlcA moiety could reorient the phosphate group relative to its position in the crystal structures, potentially altering the lysine–phosphate electrostatic interactions.

Glycosyltransferases involved in xanthan-gum biosynthesis, such as GumI (GT94) and GumH (GT4), utilize Man–GDP instead of GlcA–UDP [59,60]. Structural alignment shows that, unlike GumK, these enzymes lack Lys307 but feature a lysine–arginine pair on the opposite side (Fig 3 and S17 Fig). In GT4 enzymes, this Arg–Lys pair coordinates the diphosphate group and orients the donor substrate [61,62]. GumK lacks this motif, but Arg29 in the acceptor domain may partially compensate by interacting with the phosphate or carboxylate groups. Additionally, the position corresponding to

Lys307 contains a neutral residue in GumH- and GumI-like enzymes (S17 Fig), suggesting that the polarity of the sugar-binding site may influence selectivity.

MD simulations by Salinas *et al.* [18] did not detect an interaction between the GlcA carboxylate and Lys307, although they observed an interaction with Arg29. However, mutating this residue to Ala did not affect enzyme activity.

HREX simulations, performed with restraints on the UDP moiety and a funnel-shaped potential on the sugar diphosphate, revealed distinct GlcA–UDP binding modes (Fig 6). The main clusters show that GlcA–UDP constrains phosphate positioning toward Arg29, potentially stabilizing the interdomain salt bridge and promoting a closed conformation. Similar substrate-induced stabilization of closed states has been observed in other bacterial glycosyltransferases, where the nucleotide region regulates domain interactions and conformational transitions [61,62]. By contrast, Glc–UDP is more flexible and lacks a direct C6–Lys307 interaction, likely reducing its ability to promote domain closure. These structural distinctions between an active substrate and a structurally similar but non-reactive ligand highlight the key determinants of substrate specificity in GumK.

Our HREX simulations and bioinformatic analysis suggest two hypotheses for GlcA–UDP specificity in GumK. The first proposes that the carboxylate–Lys307 interaction is part of the binding pathway, enhancing stability and helping to orient the diphosphate toward Arg29 to promote closure. This does not exclude Glc–UDP activity but implies a higher $K_M$ due to weaker stabilization. The second hypothesis suggests that this interaction is critical for catalysis, orienting the sugar ring and neutralizing a negative charge that could interfere with the nucleophilic attack of the acceptor substrate. If correct, Glc–UDP would be a poor substrate or exhibit significantly lower catalytic efficiency (Fig 6). No *in vitro* assays have yet tested GumK activity on alternative donor substrates, leaving its specificity unresolved. Experimental validation is needed to determine whether GumK can utilize Glc–UDP or is strictly limited to GlcA–UDP.

## Conclusion

In this study, we computationally characterized GumK, a glycosyltransferase involved in xanthan-gum biosynthesis, by exploring its conformational landscape, substrate interactions, and membrane association. Our findings reveal a complex interplay between structural flexibility and membrane anchoring, which appears to be a shared feature among membrane-associated GTs and may represent a generalizable regulatory mechanism within this enzyme class.

We identified that GumK's acceptor substrate binds within a well-defined clamp in the acceptor domain, where it plays a critical role in modulating enzyme conformational dynamics. Acceptor binding enhances domain opening and destabilizes interdomain interactions, whereas donor-substrate binding induces domain closure, promoting functional cycling. This substrate-driven conformational equilibrium provides a mechanistic framework for understanding how glycosyltransferases regulate catalysis through structural rearrangements. Moreover, our results suggest that GumK selectively accommodates acceptor substrates with lipid carriers of up to approximately three isoprenyl units, which may guide future efforts in designing artificial substrates for biochemical assays and crystallization studies.

At the donor-binding site, our simulations indicate that specificity for glucuronate–UDP arises from a conserved electrostatic environment that stabilizes the substrate and facilitates domain closure. This charge-based recognition mechanism differentiates GumK from homologous enzymes such as GumI and GumH, where variations in charge distribution correspond to distinct substrate preferences. These findings provide a broader perspective on the selectivity of GT-B glycosyltransferases, underscoring the role of electrostatic interactions and conformational plasticity in substrate recognition.

A key outcome of this work is the identification of functionally relevant residues that contribute to GumK's catalytic cycle and substrate binding. A detailed summary of these residues and their proposed functional roles is provided in S1 Table, which serves as a reference for future experimental validation and biochemical characterization.

From a computational standpoint, this study highlights the challenges in sampling the full conformational space of GumK due to its intrinsic flexibility and membrane interactions. Although principal component analysis provided meaningful pathways for describing domain motions, future enhanced-sampling strategies could further elucidate how substrate

binding influences enzyme dynamics. Additionally, the identification of key conformational states in our work suggests potential collective variables (CVs) for free-energy calculations aimed at quantifying GumK's conformational transitions.

Several limitations of the computational approach still require experimental evaluation. Among these, the intrinsic complexity of GumK's conformational space and that of its substrates makes achieving full convergence challenging, even with the approximations and constraints employed. Furthermore, classical force fields cannot fully capture the complex chemistry of active carbohydrates, limiting the reliability of converged populations even when attainable [25].

Overall, our findings enhance the molecular understanding of GumK's function and substrate specificity and provide a conceptual and computational framework for studying the broader GT-B enzyme family. These insights lay the foundation for future experimental investigations and enzyme-engineering efforts aimed at developing glycosyltransferases with tailored substrate specificities.

## Supporting information

**S1 Fig. Quality assessment of the cylindrical-restrained OPES simulation using the protein–protein rescaled-interaction MARTINI force field.** A) Time evolution of the tilt angle of the acceptor domain relative to the membrane. Multiple transitions are sampled during the simulation. B) Time evolution of the distance between the acceptor domain and the membrane. Several transitions are observed, and when the distance falls below 4 nm, the protein is considered bound. C) Three-dimensional reweighted free-energy surface (FES) showing two main minima. D) Schematic representation of the cylindrical restraint. The black cylinder illustrates a $z$-aligned positional restraint applied to a reference atom in the acceptor domain. The membrane distance $d_A$ is the projection of this atom onto the cylinder axis, while the tilt angle $\tau$ is measured relative to the membrane normal ($z$-axis) using a second acceptor-domain atom to define the internal vector. (EPS)

**S2 Fig. Quality analysis of the Hamiltonian replica exchange simulation used to sample glucuronate-UDP bound to the GumK donor domain.** A) Probability matrix of the 20 replicas involved in the simulation. The replicas generally diffuse along the ladder, although some remain trapped in specific positions. B) Mean square displacement of the replicas in replica space. This indicates that while the replicas diffuse well along the ladder over certain periods, the system does not reach full stationarity, suggesting that the simulation is not fully converged. C) Distribution of the distance between the sugar C6 atom and the $N^\zeta$ atom of Lys307. The results show that the replicas sample two main states: one where the carboxylate group interacts with Lys307 (below 0.6 nm) and another where the phosphates interact with the carboxylate. D) Definition of the polar space used to describe the orientation of the sugar ring relative to the protein. The normal vector of the sugar ring is computed and translated to the origin of the inertia axes of the $\alpha$-carbons in the initial donor-domain conformation. The entire trajectory is first aligned to this structure to ensure that the inertia axes remain consistently aligned across all frames. The polar angles of the translated normal vector are then used to describe the orientation of the sugar ring. The distance between C6 and Lys307 is also shown. (EPS)

**S3 Fig. Quality analysis of the Hamiltonian replica exchange simulation for glucose-UDP bound to the GumK donor domain.** A) The probability matrix shows good mixing of the replicas, with only a few remaining trapped in specific positions along the replica ladder. B) Mean square displacement of the replicas in replica space shows stable behavior, with replicas diffusing efficiently. However, a decrease in the trend at the end of the simulation suggests that full convergence may not have been reached. C) Distance distribution between the sugar C6 atom and the $N^\zeta$ atom of Lys307. The distribution is multimodal but consistently centered around 0.8 nm across all 18 replicas. This distance is compatible with the phosphate groups interacting with Lys307. The same definitions of the collective variables (CVs) as in S2 Fig are used in this case. (EPS)

**S4 Fig. Square fluctuations of the first four normal modes derived from the normal-mode analysis of the GumK crystal structure.** A) The first normal mode (NM1) involves coordinated movements across different regions of both domains, primarily corresponding to a twisting motion. B) The second normal mode (NM2) exhibits smaller squared displacements compared to NM1 but still engages both domains, mainly reflecting a bending motion. C) The third normal mode (NM3) shows squared displacements similar to those of NM2, with motions becoming more localized within each domain. This mode helps define the opening path described in the main text. D) The fourth normal mode (NM4) is highly localized within the acceptor domain, indicating that it primarily describes local backbone vibrations rather than large-scale conformational changes.
(EPS)

**S5 Fig. A) Eigenvalues of the first 20 normal modes computed from the normal-mode analysis of the GumK crystal structure (PDB: 2HY7).** Several discontinuities are observed, but the first two modes are closely clustered, whereas the third mode begins to exhibit a higher frequency. B) Eigenvalues of the first 20 normal modes of the open conformations sampled by ClustENMD. The first three normal modes approach zero, suggesting nearly free relative motion of the two domains.
(EPS)

**S6 Fig. Sampled conformations from the ClustENMD simulation.** A) An overtwisted conformation located at the periphery of the ensemble in the 3D normal-mode space. The donor domain is rotated by 180° around the main axis due to the high flexibility of the yellow loops. B) A metastable state positioned at positive values of normal mode 3. Helix $C\alpha 4$ occupies the acceptor substrate-binding site, likely representing an inactive conformation that is disfavored when the acceptor substrate is bound.
(EPS)

**S7 Fig. Analysis of the AlphaFold-predicted structure of GumK.** (A) The structure is colored according to the pLDDT score, showing low confidence only in the terminal regions. (B) Superposition of the AlphaFold model with the crystal structure (PDB ID: 2HY7), showing that both the backbone and the side chains closely match the experimental structure, with RMSD values of 0.4 Å and 0.5 Å, respectively.
(EPS)

**S8 Fig. Variance of the first 20 principal components from the essential dynamics analysis of the unbiased MD simulation of GumK in the membrane-bound state.** The first five principal components are sufficient to define the essential subspace, as the individual variance decreases sharply after the fifth component and approaches an asymptotic value. The cumulative variance shows that the first few principal components account for a significant percentage of the total variance.
(EPS)

**S9 Fig. Probability distributions of the trajectory frames projected onto the first eight principal components (PCs).** The first principal component exhibits a well-defined bimodal distribution, suggesting a possible transition pathway. In contrast, the last principal component displays a Gaussian distribution, indicating that this direction behaves harmonically or corresponds to a fast, fully sampled motion. Based on these distributions and the eigenvalue analysis, seven principal components are sufficient to define the essential subspace of GumK.
(EPS)

**S10 Fig. ClustENMD ensemble exploring a region near the GumK crystal structure using a target RMSD of 0.5 .** The distribution in the space defined by normal modes 2 and 3 shows that the protein follows a path close to that defined

by the first principal component from the essential dynamics analysis of the unbiased MD simulation. The red arrow represents the projection of the first principal component onto the normal mode space.
(EPS)

**S11 Fig. A) Bootstrap-normalized block covariance overlap of a 1 µs simulation of GumK in the membrane-bound state.** The curve is expected to decay exponentially toward 1 as the essential subspace becomes fully sampled. B) Exponential fit of the decay, indicating that at least 3 µs of simulation are required to achieve satisfactory convergence.
(EPS)

**S12 Fig. Overlap analysis of the first and second principal components (PCs) with their final counterparts at different time points.** After 400 ns, when partial opening is sampled, the directions of the two PCs show good overlap with the final ones.
(EPS)

**S13 Fig. Distribution of hydrophobic interaction conformations in the GumK donor domain for the four clusters sampled during the Hamiltonian replica exchange simulation of glucose-UDP (Fig 6 in the main text).** In all cases, the hydrophobic interactions are more dynamic than those observed for UDP-glucuronate (S15 Fig).
(EPS)

**S14 Fig. A) Probability density of glucuronate-UDP as a function of three collective variables (CVs): the distance between the C6 atom of the sugar and the $N^\zeta$ atom of Lys307, defined as $d_k$, and the two angles $\gamma$ and $\theta$ describing the orientation of the sugar ring in a polar coordinate system.** B) RMSD of the probability density with respect to the initial state, calculated after 1 ns of simulation. The density of states is complex, and stable populations are observed at the end of the simulation. C) Probability density of glucose-UDP. D) RMSD of the density of states calculated as in B). In this case, convergence is even clearer. The density functions were computed by including all replicas from the Hamiltonian replica exchange simulation, with weights determined using the WHAM algorithm.
(EPS)

**S15 Fig. Distribution of hydrophobic interaction conformations in the GumK donor domain for the three clusters sampled during the Hamiltonian replica exchange simulation of glucuronate-UDP.** (A–C) Panels A, B, and C correspond to clusters A, B, and C, respectively, as defined in Fig 6 in the main text. Cluster A exhibits a well-localized distribution, with the residues consistently remaining in contact. In contrast, clusters B and C display more dynamic interactions, showing greater variability in residue contacts. (D) Definition of the two distances used to characterize these distributions.
(EPS)

**S16 Fig. Distribution of hydrophobic interaction conformations in the GumK donor domain for the four clusters sampled during the Hamiltonian replica exchange simulation of glucose-UDP (Fig 6 in the main text).** In all cases, the hydrophobic interactions are more dynamic than those observed for glucuronate-UDP (S15 Fig).
(EPS)

**S17 Fig. Structural alignment of the regions of GumI and GumH that are expected to interact with the sugar portion of the donor domain, i.e., Mannose-GDP.** The color scheme follows the same convention as in GumK (Fig 3 in the main text). The distribution of positive charges is reversed compared to GumK, with two positive charges present in the purple region, which is neutral in GumK. Additionally, the green region is shorter than in GumK, where it includes a loop with a highly variable sequence between residues Ala293 and Pro298.
(EPS)

**S18 Fig. Starting conformation of the GumK–acceptor complex.** The substrate was manually positioned near the binding site based on insights obtained from the unbiased MD simulations of the free protein in solution and from previous test simulations.
(EPS)

**S19 Fig. (A, B) Projections of replicas 1 and 2 onto the first two principal components.** In the presence of the acceptor substrate, GumK undergoes conformational changes, although not to the same extent as in the main simulation (see main text). (C, D) Distance between the reactive oxygen atom of the acceptor substrate and the catalytic residue Asp157. In these replicas, the acceptor substrate does not establish stable interactions as observed in the main simulation, which explains why the conformational space of the protein appears more restricted.
(EPS)

**S20 Fig. Distance distribution of the reactive hydroxyl group of the acceptor substrate relative to the catalytic residue Asp157.** Below, the Ramachandran plots show the distributions of the two glycosidic bonds in the acceptor substrate of GumK (Fig 1B) sampled during the simulation. The red color represents the bound state. The distributions indicate that when the sugar is bound, it is constrained to a specific conformation corresponding to the fully extended state. The blue dots represent the glycosidic bond between the first two glucose units in the acceptor substrate. This bond is more localized than the glycosidic bond between the second glucose and mannose, suggesting that in the bound state, the first unit is more restricted, while the second remains more flexible as it is exposed to the cleft between the two domains.
(EPS)

**S21 Fig. A) OPM prediction of the GumK–membrane complex with the protein in an close conformation.** Helix N$\alpha$4 is inserted into the membrane through its two tryptophan residues. B) OPM prediction of the open conformation of GumK interacting with the membrane. The conformation was taken from ClustENMD and shows a binding mode similar to panel (A), except for minor differences due to the distinct conformation of the clamp.
(EPS)

**S1 Table This table reports the functional classification of the residues identified in the simulation.**
(DOCX)

## Author contributions

**Conceptualization:** Davide Luciano, Gaston Courtade.

**Formal analysis:** Davide Luciano.

**Funding acquisition:** Gaston Courtade.

**Investigation:** Davide Luciano, Gaston Courtade.

**Methodology:** Davide Luciano, F. Emil Thomasen, Kresten Lindorff-Larsen.

**Project administration:** Gaston Courtade.

**Resources:** Kresten Lindorff-Larsen, Gaston Courtade.

**Supervision:** Kresten Lindorff-Larsen, Gaston Courtade.

**Visualization:** Davide Luciano.

**Writing – original draft:** Davide Luciano, Gaston Courtade.

**Writing – review & editing:** Davide Luciano, F. Emil Thomasen, Kresten Lindorff-Larsen, Gaston Courtade.

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
