## [Decision Letter · Decision Letter 0]

12 Oct 2025

PCOMPBIOL-D-25-01831

Computational characterization of the xanthan gum glycosyltransferase GumK

PLOS Computational Biology

Dear Dr. Courtade,

Thank you for submitting your manuscript to PLOS Computational Biology. After careful consideration, we feel that it has merit but does not fully meet PLOS Computational Biology's publication criteria as it currently stands. Therefore, we invite you to submit a revised version of the manuscript that addresses the points raised during the review process.

Please submit your revised manuscript within 60 days Dec 12 2025 11:59PM. If you will need more time than this to complete your revisions, please reply to this message or contact the journal office at ploscompbiol@plos.org. Please include the following items when submitting your revised manuscript:

We look forward to receiving your revised manuscript.

Kind regards,

Jianhan Chen, Ph.D.

Academic Editor

PLOS Computational Biology

Arne Elofsson

Section Editor

PLOS Computational Biology

**Additional Editor Comments :**

There is a strong recognition of the importance of the topic and the strength of integrating MD and bioinformatics. Yet, there are a range of concerns identified regarding various technical aspects of this work. Please thoroughly address these concerns in the revised manuscript, which will be re-assessed by the reviewers. Thanks.

**Journal Requirements:**

At this stage, the following Authors/Authors require contributions: Davide Luciano, F. Emil Thomasen, Kresten Lindorff-Larsen, and Gaston Courtade. Please ensure that the full contributions of each author are acknowledged in the "Add/Edit/Remove Authors" section of our submission form.

2) If any authors received a salary from any of your funders, please state which authors and which funders.

6) Please revise your current Competing Interest statement to the standard "The authors have declared that no competing interests exist."

**Reviewers' comments:**

Reviewer's Responses to Questions

Reviewer #1: The authors reported their computer simulation findings for GumK, a glycosyltransferase from Xanthomonas campestris (a Gram negative bacteria) required for xanthan gum biosynthesis. The authors performed a series of normal mode analyses, CG metadynamics, all-atom MD, Hamiltoniam REMD, and bioinformatics sequence analyses to provide potential mechanistic insight and conformational dynamics behavior for GumK. Given the industry application and its pathogenicity to plants, the study is of general scientific interest. The authors provided a systematic study with incremental complexity in modeling and simulating the putative mode of membrane/substrate binding, and protein dynamics to elucidate its enzymatic function. There are some major concerns about the simulated systems I would like the authors to address prior to further consideration for publication.

Major:

Why did the authors use a less accurate, predicted AlphaFold structure as a starting point for CG metadynamics MD simulation when a high resolution of X-ray structures available (resolution for unbound GumK is 1.9 angstrom)? Did the author perform structure comparison analysis between the predicted and X-ray structure. While AlphaFold is well suited in predicting the 3D fold of the structure based on AI training from previously solved structure, it cannot predict structures on the unsolved N and C terminus of GumK and has problems in predicting the sidechain packing of the protein.

How did the authors “docked” the GumK to the membrane? From Figure 3B, it seems a portion of the N-terminal domain was inserted into the membrane. What was the rationale behind it? Was it based primarily on Fig 5 illustration from ref 12?

While ref 12 stated that GumK is a membrane-associated enzyme, and provided a putative mode of binding to the membrane, the region shown is highly hydrophilic with large number of positively charged residues, making membrane insertion highly unlikely (Fig 3). Was that the reason why the authors had to rescaled the protein-protein interaction and apply cylindrical-shaped restraining potential to maintain a stable docked complex within the membrane? As stated by the authors, the OPES “simulation is not fully converged” after 5 us of simulation. That is a major concern. What happens when these constraints were removed? Did the CaRMSD of the -terminal region collapse, or did it flew out of the membrane?

Membrane-associated does not imply membrane-inserted. There is lack of strong support for the proposed membrane bound mechanism in ref 12 Fig 5. Given the acceptor substrate (Man-Cel-UNDDP) consists of a long unsaturated aliphatic chain, it is more likely the membrane-association of GumK is due to its binding to the membrane inserted acceptor substrate. This would allow the cationic site in the N-terminal domain be placed directly above the zwitterionic headgroup of the membrane lipids for favorable electrostatic complementary interactions.

Please provide a figure of the acceptor-protein complex used for the all atomistic unbiased MD simulation. How was the acceptor substrate docked into the protein-membrane complex. Was the simulation done with both donor and acceptor substrate. It was not clear from the method section. Also, was the complex system model based on the endpoint from the meta dynamics simulation? The bound structure of GumK (PDB: 2Q6V) only has the donor substrate? Modeling of the acceptor substrate into glycosyl transferase has been a challenge to many previous simulation studies.

Why did the authors perform NPT instead of the typical NPAT simulation for membranes?

Using artificial constraints to maintain protein binding to membrane is not a strong scientific support for the hypothesized membrane bound complex from ref 12.

The authors stated that three additional MD simulation of GumK-acceptor complex in membrane. Please include that in the method section to prevent the reader from guessing what protein-substrate-membrane bound complex the authors are referring to.

Contrary to the three stated arguments, the authors used a predicted, not experimental structure in the metadynamics simulation.

Previous studies have shown unbound and “closed” glycosyltransferase can undergo open-close conformations during MD simulation. It is essential for the N and C terminal domains of the unbound enzyme to "open" for substrate binding. The rationale behind using "closed" conformation avoids sampling interdomain dynamics may be flawed.

It is the “opened” conformation that enable the modeling of the bound substrate into glycosyltransferase. The potential that restrained the unbound “closed” conformation would not favor the sampling of the bound modes, as stated by the authors.

Minor:

Please specify exactly what system was used in each of the method section. In NMA, I was not sure whether the authors perform the analysis on the unbound or bound crystal structure of GumK until reading the results section.

Please confirm that the restraints were removed in the production run in the unbiased MD simulation.

Reviewer #2: This manuscript provides a very comprehensive computational study of the GT-B glycosyltransferase GumK, a key enzyme in xanthan gum biosynthesis. The authors employ multi-scale simulation approaches (NMA, ClustENMD, coarse-grained and all-atom MD, HREX) combined with bioinformatic analysis to probe the enzyme’s conformational dynamics, membrane interaction, and substrate specificity. The work addresses an important and underexplored area in glycosyltransferase biology. The findings on the substrate-dependent catalytic cycle (acceptor binding promotes opening, donor binding stabilizes closure) and the identification of key residues for specificity are novel and significant. However, the manuscript requires revision to improve before it can be recommended for publication.

1. The authors utilize different force fields (CHARMM36m for unbiased MD/HREX, AMBER99SB-ILDN in ClustENMD). A clear justification for this choice is absent. The reader is left wondering if this was due to technical constraints of the different software packages or a conscious choice based on the specific strengths of each force field for the different types of simulations. The authors should briefly explain this strategy in the Methods section to preempt concerns about consistency.

2. The conformational ensemble from ClustENMD (Fig. 2) is a key starting point for the study. Are these conformations stable, or are they transient states sampled due to the specific constraints of the ClustENMD method? The authors should strengthen their argument by: Explicitly stating whether any of the major conformational states (especially the open states) were observed and were stable in the subsequent, more physiologically relevant, unbiased MD simulations of the membrane-bound protein (the 1 µs trajectory). If not observed in the unbiased MD, the authors should discuss the limitations of the ClustENMD approach in the context of a membrane-anchored protein and temper their conclusions accordingly.

3. The manuscript employs a large number of simulations with different system setups (protein alone, protein+membrane, protein+donor, protein+acceptor, etc.). A table summarizing all simulations performed would be immensely helpful. This table should include: Simulation Type (e.g., Unbiased MD, HREX), System Description (e.g., Full GumK + membrane, Donor Domain + GlcA-UDP), Force Field, Simulation Time, and Key Objective. This would greatly enhance the readability and allow readers to easily grasp the full scope of the computational work.

4. The manuscript requires thorough proofreading to correct typographical and grammatical errors. Please see lines 170 and 233. The text contains several other minor errors that should be corrected.

**Have the authors made all data and (if applicable) computational code underlying the findings in their manuscript fully available?**

Reviewer #1: Yes

Reviewer #2: Yes

PLOS authors have the option to publish the peer review history of their article (what does this mean?). If published, this will include your full peer review and any attached files.

Reviewer #1: No

Reviewer #2: No

**Figure resubmission:**
---

## [Decision Letter · Decision Letter 1]

1 Dec 2025

Dear Courtade,

We are pleased to inform you that your manuscript 'Computational characterization of the xanthan gum glycosyltransferase GumK' has been provisionally accepted for publication in PLOS Computational Biology.

Best regards,

Jianhan Chen, Ph.D.

Academic Editor

PLOS Computational Biology

Arne Elofsson

Section Editor

PLOS Computational Biology

Reviewer's Responses to Questions

**Comments to the Authors:**

Reviewer #1: The authors have address all my concerns. Acceptance for publication is recommended.

Reviewer #2: No further comments

**Have the authors made all data and (if applicable) computational code underlying the findings in their manuscript fully available?**

Reviewer #1: Yes

Reviewer #2: Yes

PLOS authors have the option to publish the peer review history of their article (what does this mean?). If published, this will include your full peer review and any attached files.

Reviewer #1: No

Reviewer #2: **Yes: **Yibo Wang

---

## [Editor Report · Acceptance letter]

PCOMPBIOL-D-25-01831R1

Computational characterization of the xanthan gum glycosyltransferase GumK

Dear Dr Courtade,

I am pleased to inform you that your manuscript has been formally accepted for publication in PLOS Computational Biology. Your manuscript is now with our production department and you will be notified of the publication date in due course.

With kind regards,

Olena Szabo
